# Sound Logical Explanations for Mean Aggregation Graph Neural Networks

**Matthew Morris**
Department of Computer Science
University of Oxford
matthew.morris@cs.ox.ac.uk

**Ian Horrocks**
Department of Computer Science
University of Oxford
ian.horrocks@cs.ox.ac.uk

## Abstract

Graph neural networks (GNNs) are frequently used for knowledge graph completion. Their black-box nature has motivated work that uses sound logical rules to explain predictions and characterise their expressivity. However, despite the prevalence of GNNs that use mean as an aggregation function, explainability and expressivity results are lacking for them. We consider GNNs with mean aggregation and non-negative weights (MAGNNs), proving the precise class of monotonic rules that can be sound for them, as well as providing a restricted fragment of first-order logic to explain any MAGNN prediction. Our experiments show that restricting mean-aggregation GNNs to have non-negative weights yields comparable or improved performance on standard inductive benchmarks, that sound rules are obtained in practice, that insightful explanations can be generated in practice, and that the sound rules can expose issues in the trained models.

## 1 Introduction

Knowledge graphs (KGs) [17] find use in a number of applications [16, 41, 45]. However, KGs are often incomplete, creating the need for models to predict their missing facts. Neural models such as graph neural networks (GNNs) [13] are frequently used for this task [18, 22, 25], as well as a variety of others, including predicting properties of drug combinations [9] and recommender systems [11].

However, the predictions of neural models cannot easily be explained and verified [12]. In contrast, logic-based and neuro-symbolic methods for KG completion often yield logical rules which can be used to explain predictions [24, 29, 31, 46]. To ensure that the rules truly express the reasons why the model makes a particular prediction, it is important to ensure that they are *sound*, in the sense that applying the rules to an arbitrary dataset produces only facts that are predicted by the model. Thus, there is growing interest in models whose predictions can be characterized by sound rules [36, 42].

When it comes to GNNs, Tena Cucala et al. [35, 37] provide sound Datalog rules and equivalent programs for GNNs with non-negative weights and max or sum aggregation, respectively. Morris et al. [25] consider GNNs with sum aggregation and show that, in practice, they often provably have no sound Datalog rules. There is also a plethora of related work considering the logical expressivity of general aggregate-combine GNNs. Barceló et al. [7] show that if a GNN captures a rule, then it can be expressed in the logic ALCQ. Other works provide GNN expressivity results for various extensions of first-order logic, including linear programming [26], Presburger quantifiers [8], and counting terms [14]. Ahvonen et al. [3], Pflueger et al. [27] characterise recurrent GNNs using fixpoint operators. Cuenca Grau et al. [10] consider GNNs with bounded aggregation functions and show their equivalence to fragments of first-order logic.

Although any function mapping multisets of real numbers to real numbers can be used for the aggregation function in a GNN, standard options include mean, sum, max, and attention [40]. In practice, mean aggregation often emerges as the default choice [16, 19, 30, 32, 44] due to its

simplicity, stability during training, and good empirical performance. With regard to the expressivity of mean-GNNs, Rosenbluth et al. [30] study mean, sum, and max aggregation in terms of their ability to approximate one another. However, to our knowledge, there are no prior works analysing the logical expressivity of or attempting to extract sound rules from GNNs that use mean aggregation (mean-GNNs).

**Our Contribution** To obtain explanatory sound rules for mean-GNNs, we consider mean-GNNs with non-negative weights (MAGNNs). This isolates the source of the non-monotonicity to the aggregation function (since negative weights can also be a source of non-monotonicity), enabling a simpler analysis. This extends the work of Tena Cucala et al. [37], who proved that GNNs with non-negative weights and max or sum aggregation are *monotonic under injective homomorphisms*, enabling the verification of sound rules, and of Tena Cucala et al. [35] and Morris et al. [25], who found that restricting GNNs to non-negative weights did not greatly impact performance when using max or sum aggregation, respectively.

We provide two main theoretical results. First, we prove the exact class of "monotonic rules" that can be sound for MAGNNs, which turns out to be very limited. We also provide a means for checking if such rules are sound. Second, in light of the fact that some MAGNNs do not have equivalent first-order logic (FOL) programs, we instead provide a restricted fragment of FOL that contains sound rules that can be used to explain any prediction of a MAGNN. In our experiments, we show that mean-GNNs still perform well on benchmark datasets when restricted to having non-negative weights. We also demonstrate that they recover a variety of sound monotonic rules, as well as providing examples of the generated rules that explain predicted facts of the GNN. Despite good model performance on the test set, we find that MAGNNs still learn some nonsensical sound rules, showing the importance of explainability.

## 2 Background

**Datasets and Graphs** We fix a signature of countably infinite, disjoint sets of unary/binary predicates and constants. We also consider a countably infinite set of variables disjoint with the sets of predicates and constants. A term is a variable or a constant. An atom is an expression of the form $R(t_1, t_2)$ or $U(t_1)$, where each $t_i$ is a term and $R, U$ are binary and unary predicates, respectively. An atom is ground if it contains no variables. A fact is a ground atom and a dataset $D$ is a finite set of facts. We denote the set of all constants in $D$ with $\mathtt{con}(D)$.

For $\mathbf{v}$ a vector and $i > 0$, $\mathbf{v}[i]$ denotes the $i$-th element of $\mathbf{v}$ (likewise for matrices). For a finite set Col of colours and $\delta \in \mathbb{N}$, a (Col, $\delta$)-graph $G$ is a tuple $\langle V, \{E^c\}_{c \in \mathrm{Col}}, \lambda \rangle$ where $V$ is a finite vertex set, each $E^c \subseteq V \times V$ is a set of directed edges with colour $c$, and $\lambda$ assigns to each $v \in V$ a vector of dimension $\delta$. Each vertex $v^a \in V$ is defined to correspond uniquely to a constant $a$ from the signature. When $\lambda$ is clear from the context, we abbreviate the labelling $\lambda(v)$ as $\mathbf{v}$. We uniquely associate each $\mathbf{v}[i]$ with a unary predicate $U_i$ and each colour in $c \in \mathrm{Col}$ with a binary predicate $R^c$. There is thus a one-to-one correspondence between graphs and datasets [37]. We use $N_c(v)$ to denote the $c$-coloured neighbours of a vertex $v$. Graph $G$ is undirected if $E^c$ is symmetric for each $c \in \mathrm{Col}$ and is Boolean if $\mathbf{v}[i] \in \{0, 1\}$ for each $v \in V$ and $i \in \{1, ..., \delta\}$.

**FOL Rules** We use standard first-order logic (FOL) syntax, including equality. A variable in a FOL formula is free if it is not quantified. Barceló et al. [7] define first-order logic (FOL) classifiers over coloured graphs, where for graph $G$, node $v$, and FOL formula $\varphi$ with a free variable, $(G, v) \models \varphi$ denotes $\varphi$ being satisfied by node $v$ of $G$. Similarly, given the correspondence between graphs and datasets, for a dataset $D$, constant $a \in D$, and FOL formula $\varphi$ with free variable $x$, we let $(D, a) \models \varphi$ denote $\varphi$ being satisfied by constant $a$ of $D$. More precisely, $(D, a) \models \varphi$ if for variable substitution $\mu$ mapping $x \mapsto a$, we have $D \models \varphi\mu$. As usual in this setting, $D$ is treated as an interpretation and $D \models \varphi\mu$ iff $\varphi\mu$ evaluates to true in $D$.

We define a FOL *rule* to be of the form $B \to A(x)$, where $B$ is a FOL formula with free variable $x$ and $A$ a unary predicate. A program is a set of rules. Given a dataset $D$ and FOL rule $r : B \to A(x)$, we define the immediate consequence operator $T_r(D) = \{A(a) \mid a \in \mathtt{con}(D), (D, a) \models B\}$. For a program $\alpha$, we likewise define $T_\alpha(D) = \bigcup_{r \in \alpha} T_r(D)$. This approach is similar to that of Barceló et al. [7], except that instead of binary classification, we are essentially doing multi-class classification.

It is the same approach of Tena Cucala et al. [37], where they find equivalent programs and sound rules for different GNNs, but use Datalog semantics.

We also consider fragments of FOL defined by description logics such as ALCQ [6], the concepts of which are defined by:

$$C ::= \top \mid \bot \mid A \mid \neg C \mid C \sqcap D \mid C \sqcup D \mid \forall P.C \mid \exists P.C \mid \leq_n P.C \mid \geq_n P.C,$$

for any relation (binary predicate) $P$, ALCQ concepts $C, D$, and atomic concept (unary predicate) $A$. We define an ALCQ rule to be a subsumption of the form $C \sqsubseteq A$. Immediate consequences are defined in the same way as for FOL, since every ALCQ concept corresponds to a FOL formula with one free variable.

**Graph Neural Networks**  A function $\sigma : \mathbb{R} \to \mathbb{R}$ is *monotonically increasing* if $x < y$ implies $\sigma(x) \leq \sigma(y)$. We apply functions to vectors element-wise. A (Col,$\delta$)- graph neural network $\mathcal{M}$ with $L \geq 1$ layers is a tuple

$$\langle\, \{\mathbf{A}_\ell\}_{1 \leq \ell \leq L},\ \{\mathbf{B}_\ell^c\}_{c \in \mathrm{Col}, 1 \leq \ell \leq L},\ \{\mathbf{b}_\ell\}_{1 \leq \ell \leq L},\ \{\sigma_\ell\}_{1 \leq \ell \leq L},\ \{\mathrm{agg}_\ell\}_{1 \leq \ell \leq L},\ \mathtt{cls}_t\,\rangle, \qquad (1)$$

where, for each $\ell \in \{1, \dots, L\}$ and $c \in \mathrm{Col}$, matrices $\mathbf{A}_\ell$ and $\mathbf{B}_\ell^c$ are of dimension $\delta_\ell \times \delta_{\ell-1}$ with $\delta_0 = \delta_L = \delta$, $\mathbf{b}_\ell$ is a vector of dimension $\delta_\ell$, $\sigma_\ell : \mathbb{R} \to \mathbb{R}^+ \cup \{0\}$ is a monotonically increasing continuous activation function with non-negative range, $\mathrm{agg}_\ell$ is an aggregation function from finite real multisets to real values, and $\mathtt{cls}_t : \mathbb{R} \to \{0, 1\}$ for threshold $t_\mathcal{M} \in \mathbb{R}$ is a step classification function such that $\mathtt{cls}_t(x) = 1$ if $x \geq t$ and $\mathtt{cls}_t(x) = 0$ otherwise.[1]

Applying $\mathcal{M}$ to a $(\mathrm{Col}, \delta)$-graph induces a sequence of labels $\mathbf{v}_0, \mathbf{v}_1, ..., \mathbf{v}_L$ for each vertex $v$ in the graph as follows. First, $\mathbf{v}_0$ is the initial labelling of the input graph; then, for each $1 \leq \ell \leq L$, $\mathbf{v}_\ell$ is defined by the following expression:

$$\sigma_\ell\big(\mathbf{b}_\ell + \mathbf{A}_\ell \mathbf{v}_{\ell-1} + \sum_{c \in \mathrm{Col}} \mathbf{B}_\ell^c\, \mathrm{agg}_\ell(\{\!\{\mathbf{u}_{\ell-1} \mid (v, u) \in E^c\}\!\})\big) \qquad (2)$$

The output of $\mathcal{M}$ is a (Col,$\delta$)-graph with the same vertices and edges as the input graph, but where each vertex is labelled by $\mathtt{cls}_t(\mathbf{v}_L)$.

When each $\mathrm{agg}_\ell$ is mean, we call it a mean-GNN (likewise for max and sum). When all values in each $\mathbf{A}_\ell$ and $\mathbf{B}_\ell^c$ are non-negative, we say that the GNN is *monotonic*. In this paper, we consider in particular the class of *monotonic mean GNNs* (MAGNNs). Besides the restriction of non-negative weights, this is the same model used in R-GCN [32] (where the normalisation constant is set as described in their paper, to be the number of $c$-coloured neighbours of the node).

**Dataset Transformations Through GNNs**  A GNN $\mathcal{M}$ induces a transformation $T_\mathcal{M}$ from datasets to datasets over a given finite signature [37]. To this end, the input dataset must be first encoded into a graph that can be directly processed by the GNN, and the graph resulting from the GNN application must be subsequently decoded back into an output dataset. We adopt the so-called *canonical scheme*, where $\mathtt{enc}(D)$ of a dataset $D$ is the Boolean $(\mathrm{Col}, \delta)$-graph with a vertex $v^a$ for each constant $a$ in $D$ and a $c$-coloured edge $(v^a, v^b)$ for each fact $R^c(a, b) \in D$. Furthermore, given a vertex $v^a$, vector component $\mathbf{v}^a[p]$ is set to 1 if and only if $U_p(a) \in D$, for $p \in \{1, \dots, \delta\}$. The decoder $\mathtt{dec}$ is the inverse of the encoder. The *canonical* dataset transformation induced by a GNN $\mathcal{M}$ is then defined as: $T_\mathcal{M}(D) = \mathtt{dec}(\mathcal{M}(\mathtt{enc}(D)))$. We abbreviate $\mathcal{M}(\mathtt{enc}(D))$ by $\mathcal{M}(D)$.

In this paper, we will only consider $(\mathrm{Col}, \delta)$- graphs, datasets, and GNNs, unless specified otherwise. When performing the task of link prediction, we use the additional dataset transformations of Tena Cucala et al. [37], which enable GNNs to predict binary facts.

**Soundness and Subsumption**  A FOL (or ALCQ) logic program or rule $\alpha$ is sound for a MAGNN $\mathcal{M}$ if $T_\alpha(D) \subseteq T_\mathcal{M}(D)$ for each dataset $D$. Conversely, $\alpha$ is complete for $\mathcal{M}$ if $T_\mathcal{M}(D) \subseteq T_\alpha(D)$ for each dataset $D$. We say that $\alpha$ is equivalent to $\mathcal{M}$ if it is both sound and complete for $\mathcal{M}$. Finally, we say that a rule or program $R$ subsumes a rule $r$ if for any dataset $D$, $T_r(D) \subseteq T_R(D)$.

---

[1]Note that if $>$ is used for the classifier instead of $\geq$, one obtains an entirely different set of theoretical results from the ones in this paper.

# 3  Sound Rules and Explanations

We prove a series of theoretical results for MAGNNs, to identify which monotonic rules can be sound for them and obtain sound explanatory rules. The restriction to non-negative weights isolates the source of the non-monotonicity to the aggregation function, making analysis easier; however, MAGNNs still possess the defining feature of mean-GNNs in their choice of aggregation function. This restriction is the same approach that was used effectively for GNNs with sum or max aggregation [25, 35, 37]. First, we show that there exist MAGNNs with no equivalent FOL programs. Intuitively, this follows from FOL's inability to express numerical comparisons.

**Proposition 1.** *There exists a MAGNN $\mathcal{M}$ such that for any dataset $D$ and constant $a \in \texttt{con}(D)$, $U(a) \in T_{\mathcal{M}}(D)$ if and only at least half of the neighbours $b$ of $a$ in $D$ are such that $U(b) \in D$, where $U$ is a unary predicate. This logical function cannot be defined in FOL [21].*

A full proof is given in Appendix A.1. This means that the task of trying to find equivalent programs for them, such as the approach of Tena Cucala et al. [37], is impossible. Instead, we aim to extract sound rules from MAGNNs, to explain their predictions. Thus, we consider in particular a class of "monotonic" rules encompassing many common inference patterns, and identify the restricted fragment that can be sound for MAGNNs. Finally, we define a rule language using a fragment of FOL that, for any mean-GNN prediction, contains a sound rule that entails the prediction.

## 3.1  Which Monotonic Rules can be Sound for MAGNNs

Since equivalent programs cannot be found in some cases, we instead first show which simple monotonic rules can be sound for MAGNNs. Monotonic rules appear commonly in practice [1, 23, 34], are often easily human-readable, and are easier to use for extracting sound rules.

**Definition 2.** *A rule $r$ is* monotonic *(under dataset extension) if for all datasets $D, D'$ such that $D \subseteq D'$, $T_r(D) \subseteq T_r(D')$.*

Description logics, such as ALCQ [6], are a natural choice for providing a language for sound rules due to their widespread use in data modelling and their theoretical relationship to GNNs [7]. However, some ALCQ concepts are inherently non-monotonic: for example, if $(D, c) \models \forall P.A$ for some atomic concept $A$, binary predicate $P$, dataset $D$, and constant $c \in \texttt{con}(D)$, then for $D' = D \cup \{P(c, d)\}$ with a fresh constant $d$, $(D', c) \not\models \forall P.A(c)$.

EL [6] is a more restricted description logic than ALCQ, where concepts use only intersection and existential quantification—any EL rule is thus monotonic. We define ELUQ, a language in-between EL and ALCQ, as the language containing concepts defined by

$$C ::= \top \mid A \mid C \sqcap D \mid C \sqcup D \mid \exists P.C \mid \geq_n P.C,$$

where $C, D$ are ELUQ concepts, $P$ is a binary predicate, $A$ is an atomic concept, and $n$ a positive integer. Note that any concept $\exists P.C$ can be written equivalently as $\geq_1 P.C$. An ELUQ rule has the form $C \sqsubseteq A$. This language is similar (but not equivalent) to that of *Datalog with inequalities* considered by Tena Cucala et al. [37] for GNNs with sum aggregation and non-negative weights, given constraints on the inequalities and the addition of disjunction. All ELUQ rules are monotonic; however, it remains an open question as to whether this is a maximal monotonic fragment of ALCQ.

We find that for any ELUQ rule $r$ that is sound for a MAGNN $\mathcal{M}$, there exists a set of rules of a very simple form that (1) are each sound for $\mathcal{M}$ and (2) collectively subsume $r$. This is formalised in the following theorem.

**Theorem 3.** *Let $\mathcal{M}$ be a MAGNN and $r$ an ELUQ rule that is sound for $\mathcal{M}$. Then there exists a finite set of rules $R$ such that:*

1. *Each $r' \in R$ has the form $\exists P_1.\top \sqcap ... \sqcap \exists P_j.\top \sqcap A_1 \sqcap ... \sqcap A_k \sqcap \top \sqsubseteq A_{k+1}$, where each $P_i$ is a binary predicate, $A_i$ a unary predicate, and $j, k \in \mathbb{N}_0$.*

2. *Each $r' \in R$ is sound for $\mathcal{M}$.*

3. *$R$ subsumes $r$.*

The proof provides a construction of $R$. For example, given an ELUQ rule $r : \geq_3 P_1.(A_1 \sqcup A_2) \sqcap \exists P_2.(\exists P_1.A_3) \sqcup A_4 \sqsubseteq A_5$ be sound for $\mathcal{M}$, we can construct rules $r_1 : \exists P_1.\top \sqcap \exists P_2.\top \sqsubseteq A_5$ and $r_2 : A_4 \sqsubseteq A_5$ such that both $r_1$ and $r_2$ are sound for $\mathcal{M}$ and $\{r_1, r_2\}$ subsumes $r$.

As a consequence of this, when checking for sound ELUQ rules it suffices to check for rules of the above restricted form. In addition, whilst the space of all ELUQ rules is infinite, the restricted fragment we provide is finite, meaning that all sound ELUQ rules can be covered by instead iterating over a finite number of restricted rules. Finally, this raises concerns for the logical expressivity of MAGNNs, since any sound ELUQ rule is ultimately subsumed by a set of much simpler sound rules.

*Proof sketch.* The full proof of Theorem 3 is given in Appendix A.2. A different $r'$ is defined for each concept in the outer disjunction of the body of $r$. To see the crucial step of the proof, consider some rule $r$ without disjunction in the outer concept, and let $r_1 := r$. Then $r_1$ is of the form $\geq_n P.C_1 \sqcap C_2 \sqsubseteq A$, where $P$ is a binary predicate, $C_1$ an ELUQ concept, $C_2$ an ELUQ concept without disjunction, and $n$ a positive integer. We inductively define a sequence $r_2, ..., r_k$ of ELUQ rules that such for all $i \in \{1, ..., k-1\}$, $r_{i+1}$ is sound for $\mathcal{M}$ and $r_{i+1}$ subsumes $r_i$. Given $r_i$ of the form $\geq_n P.C_1 \sqcap C_2 \sqsubseteq A$, we define $r_{i+1}$ as the rule $\exists P.\top \sqcap C_2 \sqsubseteq A$. Trivially, $r_{i+1}$ subsumes $r_i$.

Now consider the following datasets $D_1$ and $D_m$ (parametrised by $m \in \mathbb{N}$). In $D_1$, a concept $C_2$ annotating a constant $a$ denotes that there exists other facts in the dataset such that $(D_1, a) \models C_2$ (similarly for $D_m$).

**Dataset $D_1$:** 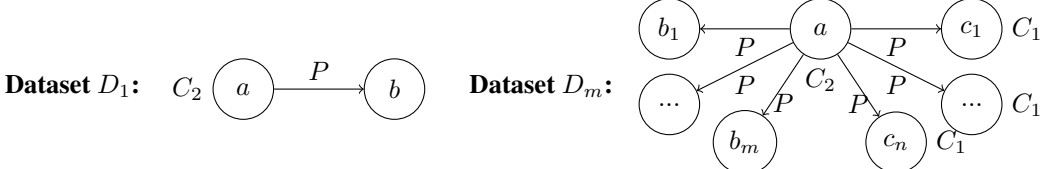 **Dataset $D_m$:**

Note that $D_1$ is an instantiation of the body of $r_{i+1}$. The soundness of $r_{i+1}$ depends on the fact that as $m \to \infty$, the computation of $\mathcal{M}$ on $D_m$ at constant $a$ tends to that of $\mathcal{M}$ on $D_1$ (and that $A(a) \in T_{r_i}(D_m)$).

Since $r_1$ contains a finite number of $\exists$ or $\geq_n$ operators and each $r_{i+1}$ has one fewer $\exists$ or $\geq_n$ operator than $r_i$, this inductive construction is guaranteed to terminate for some $k$. We include $r_k$ in $R$. Once this has been done for every $r_1 \in R'$, we have $R$ subsumes $r$. $\qquad\square$

**Checking for Monotonic Rule Soundness**    It remains to be shown how one can verify the soundness of ELUQ rules. Since we only need to consider rules of the form given in Theorem 3, there are exactly $\delta \cdot 2^{\delta \cdot |\text{Col}|}$ possible rules to check. For rules $r_1, r_2$, when considering the sets of body concepts $B_{r_1}$ and $B_{r_2}$, if $B_{r_1} \subseteq B_{r_2}$ and $r_1$ is sound, then $r_2$ is sound: we use this to optimise the procedure. The following proposition allows one to check the soundness of ELUQ rules.

**Proposition 4.** *Let $\mathcal{M}$ be a MAGNN and $r : \exists P_1.\top \sqcap ... \sqcap \exists P_j.\top \sqcap A_1 \sqcap ... \sqcap A_k \sqsubseteq A_{k+1}$ be a rule of the form given in Theorem 3. Define $D_{base} := \{A_1(a), ..., A_k(a), P_1(a, b), ..., P_j(a, b)\}$ for constants $a \neq b$. Then $r$ is sound for $\mathcal{M}$ if and only if $A_{k+1}(a) \in T_{\mathcal{M}}(D_{base})$.*

For example, to check if a rule $r : \exists P_1.\top \sqcap \exists P_2.\top \sqcap A_1 \sqcap A_2 \sqsubseteq A_3$ is sound, it suffices to compute the output of $\mathcal{M}$ on the following dataset $D_{base} = \{A_1(a), A_2(a), P_1(a, b), P_2(a, b)\}$, which can be represented as follows:

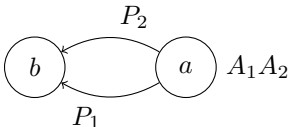

Then $r$ is sound for $\mathcal{M}$ if and only if $A_3(a) \in T_{\mathcal{M}}(D_{base})$. Note that this method also works if the body of the rule $r$ is empty, by simply checking if $A_{k+1}(a) \in T_{\mathcal{M}}(D_\emptyset)$, where $D_\emptyset = \{P(b, a)\}$.

*Proof sketch.* The full proof of Proposition 4 is given in Appendix A.3. It relies on the fact that $(D_{base}, a) \models B$, where $B$ is the body of $r$. Furthermore, no extension to $D_{base}$ will decrease the output

at $v^a$, and any dataset satisfying $B$ is an extension of $D_{\text{base}}$ (up to constant relabelling, un-merging the neighbours of $v^a$, and letting $b = a$). Note that none of these merges / un-merges will decrease the output at $v^a$ and that MAGNNs are agnostic to the particular constants used in the input dataset. □

**Consequences for Link Prediction**   To perform link prediction, we combine the canonical transformation with the dataset encoding-decoding scheme defined by Tena Cucala et al. [37], which has found wide use [22, 25, 35, 37, 43]; it introduces additional nodes in the graph to represent pairs of constants. Node labels then encode binary facts, instead of unary. Graph edges are used only to connect nodes that have constants in common; and so the presence of an edge is independent of whether any specific fact holds in the dataset.

The encoding and decoding can be described by a set of rules [37, Section 3.2], each expressed in a simple extension of Datalog. Given a sound rule for the canonical transformation, we can combine it (via unfolding) with the rules representing the encoder and decoder to obtain a Datalog rule that is sound for the entire link prediction transformation. This observation, combined with Theorem 3, ensures that each monotonic rule sound for the entire link prediction transformation is subsumed by rules of the form $R_1(x, y) \wedge ... \wedge R_{m-1}(x, y) \rightarrow R_m(x, y)$, where $R_1, ..., R_m$ are binary predicates from the signature. This is obtained by unfolding a rule of the form given in Theorem 3, since the unary predicates $A_i$ are unfolded into the binary predicates they represent, and the existentials $\exists P_j.\top$ can safely be removed since the edge presence is independent of any specific fact.

## 3.2   Explaining MAGNN Predictions

In this section, we provide a family of rules $\Omega$ (a fragment of FOL) to explain any prediction of a MAGNN. More precisely, for any MAGNN $\mathcal{M}$, dataset $D$, and predicted fact $A(a) \in T_{\mathcal{M}}(D)$, there exists a rule $r \in \Omega$ such that $A(a) \in T_r(D)$ and $r$ is sound for $\mathcal{M}$. To achieve this, we define a procedure to derive the rule $r \in \Omega$, given $\mathcal{M}$, $D$, and $A(a)$.

**Logic Language Needed**   We first define the family of rules using ALCQ syntax, with the addition of a new operator, rather than defining it directly in FOL. The operator is $\exists_n$ ("exists $n$ unique"): for example: $\exists_3 P.(A_1, A_2, \top)$. Its semantics are defined as follows. For concepts $C_1, ..., C_n$, binary predicate $P$, dataset $D$, and constant $c \in D$, we define $(D, c) \models \exists_n P.(C_1, ..., C_n)$ if and only if there exist distinct constants $c_1, ..., c_n \in \text{con}(D)$ such that $(D, c_1) \models C_1, ..., (D, c_n) \models C_n$ and $P(c, c_1), ..., P(c, c_n) \in D$. Written in first-order logic, it is defined as $(D, c) \models \exists_n P.(C_1, ..., C_n) \equiv$

$$(D, c) \models \exists x_1 ... \exists x_n (x_i \neq x_j \text{ for all } i, j \in \{1, ..., n\} \text{ such that } i < j)$$
$$\wedge\, C_1(x_1) \wedge ... \wedge C_n(x_n) \wedge P(x, x_1) \wedge ... \wedge P(x, x_n),$$

for free variable $x$. We then define the family of rules $\Omega$ as follows. An $\Omega$-concept $C$ is defined by $C ::= \top \mid A \mid C_1 \sqcap C_2 \mid \exists_n P.(C_1, ..., C_n) \sqcap\, \leq_m P.\top$, where $A$ is any atomic concept, $n$ any positive integer, $m \geq n$ an integer, $C_1, ..., C_n$ any $\Omega$-concepts, and $P$ any binary predicate. A rule in $\Omega$ has the form $C \sqsubseteq A$, where $C$ is any $\Omega$-concept and $A$ any atomic concept.

**Explanatory Rule**   Let $\mathcal{M}$ be a MAGNN, $D$ a dataset, and $A(a) \in T_{\mathcal{M}}(D)$ a prediction of the MAGNN. We define a rule $r \in \Omega$ by $C_L^a \sqsubseteq A$, where the only part of $\mathcal{M}$ that the rule depends on is $L$, which is what guarantees that the rule will be finite. The body concept $C_L^a$ is defined as follows:

**Definition 5.** *For a dataset $D$, $c \in \text{con}(D)$, $\ell \in \mathbb{N}_0$, we inductively define a concept $C_\ell^c$ by $C_\ell^c := \top \sqcap A_1 \sqcap ... \sqcap A_k$, where $A_1, ..., A_k$ are all atomic concepts $A_i$ such that $A_i(c) \in D$. Then, if $\ell = 0$, we are done: $C_\ell^c$ has been defined.*

*Otherwise, for each binary predicate $P$, let $c_1, ..., c_n$ be all the constants $c_i \in \text{con}(D)$ such that $P(c, c_i) \in D$. Then, if $n > 0$, extend $C_\ell^c$ as follows: $C_\ell^c := C_\ell^c \sqcap \exists_n P.(C_{\ell-1}^{c_1}, ..., C_{\ell-1}^{c_n}) \sqcap\, \leq_n P.\top$. $C_\ell^c$ is extended for each binary predicate $P$. Note that each $C_{\ell-1}^{c_i}$ is defined inductively.*

The following theorem then shows that the rule both explains the prediction (by producing the same fact on the dataset as the MAGNN) and is sound for the MAGNN.

**Theorem 6.** *For any MAGNN $\mathcal{M}$, dataset $D_a$, and fact $A(a) \in T_{\mathcal{M}}(D_a)$, the rule $r : C_L^a \sqsubseteq A$ (dependent on $D_a$) is sound for $\mathcal{M}$, and $A(a) \in T_r(D_a)$.*

Consider a dataset $D_a = \{A_1(a), P_1(a, b_1), P_1(a, b_2), P_2(b_3, a), A_2(b_2), P_2(b_2, c), P_3(c, d)\}$ and 2-layer GNN $\mathcal{M}$, for example, such that $A_3(a) \in T_{\mathcal{M}}(D)$. Then, using Theorem 6, we construct the rule $r : C_2^a \sqsubseteq A_3$, where $C_2^a$ is $\top \sqcap A_1 \sqcap \exists_2 P_1.(\top, \top \sqcap A_2 \sqcap \exists_1 P_2.(\top) \sqcap \leq_1 P_2.\top) \sqcap \leq_2 P_1.\top$. Notice that $(D_a, a) \models C_2^a$, so we have $A_3(a) \in T_r(D)$. Furthermore, $A_3(a) \in T_{\mathcal{M}}(D)$ is a sufficient witness for the soundness of $r$, as we will prove below.

*Proof sketch.* The full proof of Theorem 6 is given in Appendix A.5.

$(D_a, a) \models C_L^a$ follows from the inductive construction of $C_L^a$, from which we obtain $A(a) \in T_r(D_a)$. It remains to be shown that $r$ is sound for $\mathcal{M}$. Let $D_b$ be a dataset: to show soundness, we prove that $T_r(D_b) \subseteq T_{\mathcal{M}}(D_b)$, so let $A(b) \in T_r(D_b)$.

First, we define a dataset $D_L^a$ to be the $L$-hop neighbourhood of the constant $a$ in $D_a$ (analogously to how $L$-hop neighbourhoods are defined for nodes in graphs). Then we show that $A(a) \in T_{\mathcal{M}}(D_L^a)$, since $A(a) \in T_{\mathcal{M}}(D_a)$ — this follows from the definition of $D_L^a$ as the $L$-hop neighbourhood. Finally, we prove $A(b) \in T_{\mathcal{M}}(D_b)$, since $A(a) \in T_{\mathcal{M}}(D_L^a)$ — this follows from the fact that any differences between $D_b$ and $D_L^a$ cannot yield a lower MAGNN output for $v^b$ in comparison to $v^a$, plus a mapping between constants of $D_b$ and $D_L^a$. $\qquad\square$

## 4  Experiments

We train GNNs across several benchmark link prediction datasets and a node classification dataset, showing that sound rules and explanations can be found for MAGNNs in practice, and that the restriction to monotonicity does not significantly decrease performance. For the model architecture, we fix a hidden dimension of twice the input dimension, 2 GNN layers, ReLU after the first layer, and sigmoid after the second layer. The GNN definition given in Section 2, which was chosen for ease of presentation, describes GNNs aggregating in the reverse direction of the edges. For our experiments, we follow the standard approach and aggregate in the direction of the edges. Thus, when presenting a rule, we write each binary predicate as its inverse. For example, "advisor" is written as "advisorOf".

We use GNNs with max, sum, and mean aggregation. We train each model for 8000 epochs, stopping training early if loss does not improve for 50 epochs. For all trained models, we compute standard classification metrics, such as precision, recall, accuracy, and F1 score. For each model, we choose the classification threshold by computing the accuracy on the validation set across a range of 108 thresholds between 0 and 1, selecting the one which maximises accuracy. We train all our models using binary cross entropy loss and the Adam optimiser with a learning rate of 0.001. We train models without restrictions as baselines (denoted by "Standard"), as well as restricting the models to having non-negative weights (denoted by "Non-Neg") by clamping negative weights to 0 after each optimiser step, as in the approaches of Morris et al. [25], Tena Cucala et al. [35]. We run each experiment across 5 different random seeds and present the aggregated metrics. To compute 95% confidence intervals, we assume a normal distribution and compute the interval as $1.97\times$ SEM (standard error of the mean). Experiments were run using PyTorch Geometric, with 2 CPUs and 16GB of memory on a Linux server, using 34 days of compute time.

**Datasets**   We use 3 standard benchmarks: WN18RRv1, FB237v1, and NELLv1 [38], each of which provides datasets for training, validation, and testing, as well as negative examples and positive targets. Importantly, these benchmarks are also inductive, meaning that the validation and testing sets contain constants not seen during training. We also use the LUBM dataset [15, LUBM(1,0)], with the train/test split from Liu et al. [23]; this is a node classification dataset, all others are link prediction.

Finally, we utilise LogInfer [23], a framework which augments a dataset by considering Datalog rules conforming to a particular pattern and adding the consequences of the rules to the dataset. We use the datasets LogInfer-WN-hier (WN-hier) and LogInfer-WN-sym (WN-sym) [23], which are enriched with the hierarchy and symmetry patterns, respectively. We also use LogInfer-WN-hier_nmhier [25], which was created using a mixture of monotonic and non-monotonic rules: rules from the "hierarchy" and "non-monotonic hierarchy" $(R(x, y) \wedge \neg S(y, z) \to T(x, y))$ patterns, in this case. We use the dataset to test whether monotonic rules can be recovered from the dataset, despite the presence of non-monotonic rules. For the LogInfer datasets, during each training epoch, 10% of the input facts are randomly set aside and used as ground truth positive targets, whilst the rest of the facts are used as input to the model.

| Dataset | Agg | Weights | %Acc | %Prec | %Rec | F1 |
|---|---|---|---|---|---|---|
| LUBM | Mean | Standard | $97.1 \pm 0.0$ | $96.9 \pm 0.0$ | $97.2 \pm 0.0$ | $97.1 \pm 0.0$ |
| | | Non-Neg | $91.5 \pm 0.0$ | $87.8 \pm 0.0$ | $96.4 \pm 0.0$ | $91.9 \pm 0.0$ |
| WN-hier | Mean | Standard | $99.5 \pm 0.0$ | $99.5 \pm 0.0$ | $99.5 \pm 0.0$ | $99.5 \pm 0.0$ |
| | | Non-Neg | $99.8 \pm 0.0$ | $99.7 \pm 0.0$ | $99.9 \pm 0.0$ | $99.8 \pm 0.0$ |
| WN-sym | Mean | Standard | $99.4 \pm 0.0$ | $99.5 \pm 0.0$ | $99.3 \pm 0.0$ | $99.4 \pm 0.0$ |
| | | Non-Neg | $100 \pm 0.0$ | $100 \pm 0.0$ | $100 \pm 0.0$ | $100 \pm 0.0$ |
| WN-hier_nmhier | Mean | Standard | $86.2 \pm 0.0$ | $84.7 \pm 0.0$ | $88.4 \pm 0.0$ | $86.5 \pm 0.0$ |
| | | Non-Neg | $71.6 \pm 0.0$ | $80.1 \pm 0.1$ | $58.8 \pm 0.1$ | $67.2 \pm 0.0$ |
| FB237v1 | Mean | Standard | $68.7 \pm 0.0$ | $95.4 \pm 0.0$ | $39.3 \pm 0.0$ | $55.7 \pm 0.0$ |
| | | Non-Neg | $71.8 \pm 0.0$ | $75.4 \pm 0.0$ | $64.8 \pm 0.0$ | $69.7 \pm 0.0$ |
| WN18RRv1 | Mean | Standard | $93.7 \pm 0.0$ | $98.5 \pm 0.0$ | $88.8 \pm 0.0$ | $93.4 \pm 0.0$ |
| | | Non-Neg | $95.5 \pm 0.0$ | $98.1 \pm 0.0$ | $92.7 \pm 0.0$ | $95.3 \pm 0.0$ |
| NELLv1 | Mean | Standard | $75.2 \pm 0.1$ | $93.8 \pm 0.0$ | $53.4 \pm 0.2$ | $65.7 \pm 0.2$ |
| | | Non-Neg | $93.4 \pm 0.0$ | $88.8 \pm 0.0$ | $99.4 \pm 0.0$ | $93.8 \pm 0.0$ |

Table 1: Results for mean-GNNs with standard / non-negative weights, across all datasets. Metrics are computed on the test set and shown with a 95% CI.

| Dataset | Tot | Un | Bin | Mix | 0 | 1 | 2 | 3 | 4 | 5 | 6 | 7 | 8 |
|---|---|---|---|---|---|---|---|---|---|---|---|---|---|
| LUBM | 11.6 | 1.4 | 9.8 | 0.4 | 2 | 9.6 | 1.4 | 0.6 | 0 | | | | |
| WN-hier | 22.6 | 15.6 | 0 | 7 | 0 | 2.6 | 5.4 | 6.6 | 1.4 | 3.8 | 2.2 | 0.4 | 0.2 |
| WN-sym | 0.4 | 0 | 0 | 0.4 | 0 | 0 | 0 | 0 | 0.4 | 0 | 0 | 0 | 0 |
| WN-hier_nmhier | 53.6 | 4.6 | 0 | 49 | 0 | 0 | 1.8 | 16.2 | 17.8 | 12.6 | 4.8 | 0 | 0.4 |
| FB237v1 | 136 | 136 | 0 | 0 | 29 | 136 | | | | | | | |
| WN18RRv1 | 0 | 0 | 0 | 0 | 0 | 0 | 0 | 0 | 0 | 0 | 0 | 0 | 0 |
| NELLv1 | 1 | 1 | 0 | 0 | 0 | 0 | 1 | | | | | | |

Table 2: Counts of monotonic rules of the form given in Theorem 3. Tot, Un, Bin, Mix are the counts of the total number of rules, rules with only unary atoms, rules with only binary atoms, and rules with a mix of atom arities. Each remaining column $i$ counts the number of rules with $i$ body concepts.

**Rule Extraction and Explanations**  On all datasets, we use Proposition 4 to check for monotonic rules of the form given in Theorem 3. This procedure is described in Algorithm 1 of Appendix B.1. Given the differing number of unary and binary predicates in each dataset and the exponential growth of the rule space, we check all rules up to a differing number of a body concepts: 4 for LUBM, all 15 for WN-based-datasets, 1 for FB237v1, and 2 for NELLv1. This yields a total possible number of sound rules: 383670 for LUBM, all 360448 for WN18RRv1 and the LogInfer datasets, 56169 for FB237v1, and 216600 for NELLv1. When a rule is a subsumed by another with fewer body atoms, it is not counted.

We also compute sound explanatory rules of the form given in Theorem 6 for all true positive predictions made on the test set of LUBM. Given that the explanatory rules are often large, we improve the process by first checking if a sound rule of the form given in Theorem 3 derives the fact on the input dataset, and return the rule as an explanation if it does. If no such rule exists, we use rules from Theorem 6. Other strategies for improving the rule quality are discussed in Appendix B.2.

**Results**  Mean-GNN performance on the test set is shown in Table 1. For completeness, results across all aggregation functions are given in Tables 5 to 7 of Appendix C. We see similar behaviour for mean-GNNs as for sum and max-GNNs when they are restricted to non-negative weights, which matches the behaviour seen by Morris et al. [25] for sum-GNNs and Tena Cucala et al. [35] for max-GNNs. On LUBM, the restriction yields a small decrease in performance. For the monotonic LogInfer datasets, there is a slight increase in performance, and for the non-monotonic one, a substantial decrease, since the dataset explicitly penalises monotonic model behaviour. Finally, on the benchmark datasets, the restriction improves performance, sometimes substantially. This demonstrates that the

| Dataset | Examples of Sound Rules |
|---|---|
| LUBM | $\top \sqsubseteq$ Publication
$\exists$advisorOf.$\top \sqsubseteq$ AssociateProfessor
Department $\sqcap$ UndergraduateStudent $\sqcap$ $\exists$advisorOf.$\top \sqsubseteq$ FullProfessor |
| WN-hier | _member_of_domain_usage$(X, Y) \rightarrow$ _member_meronym$(X, Y)$ |
| WN-sym | _has_part$(X, Y)$ $\wedge$ _similar_to$(X, Y)$ $\rightarrow$ _derivationally_related_form$(X, Y)$ |
| WN-hier_nmhier | _hypernym$(X, Y) \wedge$ _synset_domain_topic_of$(X, Y) \rightarrow$ _also_see$(X, Y)$ |
| FB237v1 | $\top \rightarrow$ /music/instrument/instrumentalists$(X, Y)$
/base/biblioness/bibs_location/state$(X, Y) \rightarrow$ /film/film/genre$(X, Y)$ |
| NELLv1 | organizationterminatedperson$(X, Y) \wedge$ topmemberoforganization$(X, Y) \rightarrow$ organizationhiredperson$(X, Y)$ |

Table 3: Randomly sampled sound monotonic rules of the form given in Theorem 3.

| | |
|---|---|
| Fact | Publication(http://www.Department10.University0.edu/FullProfessor5/Publication3) |
| Rule | $\top \sqsubseteq$ Publication |
| Fact | ResearchAssistant(http://www.Department1.University0.edu/GraduateStudent14) |
| Rule | GraduateStudent $\sqsubseteq$ ResearchAssistant |
| Fact | GraduateStudent(http://www.Department12.University0.edu/GraduateStudent48) |
| Rule | ResearchAssistant $\sqsubseteq$ GraduateStudent |
| Fact | GraduateStudent(http://www.Department3.University0.edu/GraduateStudent44) |
| Rule | ResearchAssistant $\sqcap$ $\exists$authorOfPublication.$\top \sqsubseteq$ GraduateStudent |
| Fact | Course(http://www.Department12.University0.edu/Course1) |
| Rule | $\exists$courseTakenBy.$\top$ $\sqcap$ $\exists$hasTeacher.$\top \sqsubseteq$ Course |
| Fact | AssistantProfessor(http://www.Department13.University0.edu/AssistantProfessor10) |
| Rule | $\exists$advisorOf.$\top \sqsubseteq$ AssistantProfessor |
| Fact | University(http://www.University772.edu) |
| Rule | $\exists$grantedUndergraduateDegreeTo.$\top \sqsubseteq$ University |

Table 4: Randomly sampled explanatory rules that have successfully been reduced in size, shown with the corresponding facts produced by the MAGNN on LUBM.

restriction of mean-GNNs to MAGNNs can be done in practice without sacrificing significant model performance, while enabling the extraction of sound rules. We hypothesize that the occasional performance increase seen when restricting the models to non-negative weights is due to (1) it regularizing the model and (2) the dataset patterns not requiring negative weights to be captured.

In Table 2, we present counts of the number of sound monotonic rules that were obtained on each dataset. On LUBM, WN-hier, WN-hier_nmhier, and FB237v1, we find a number of sound rules with a varying number of body concepts. On average, we found 2 sound rules on LUBM that have empty bodies, and 29 on FB237v1; these rules are clearly absurd, but are what the GNN has learned, highlighting the need for sound rules so that such issues in the models can be exposed. Further to this, we note that almost no sound monotonic rules were obtained on WN-sym, WN18RRv1, or NELLv1. This is especially concerning for WN-sym, which was populated with the consequences of monotonic rules. This behaviour is due to the very restricted form that sound monotonic rules can have when used with the link prediction encoding of Tena Cucala et al. [37], as discussed in Section 3.1. Morris et al. [25], on the other hand, found that all of the rules used to create WN-sym were sound when using sum-GNNs with non-negative weights. We show some sample sound monotonic rules in Table 3, with more given in Table 8 of Appendix C.

In Table 9 of Appendix C, we give samples of explanatory rules of the form given in Theorem 6. On LUBM, these explanatory rules had 25 body concepts on average. When using the improved

procedure of first checking for explanatory rules of the form given in Theorem 3, the rules averaged only 11 body concepts. On average, out of 1990 true positive predictions, the process only failed 22 times to explain the predicted fact using a rule from Theorem 3. In Table 4, we give samples of such explanatory rules, which are useful for providing insights into the data, as well as exposing issues with the trained MAGNN. For example, the explanatory rule $\top \sqsubseteq$ Publication exposes a nonsensical rule learned by the MAGNN, which arises as a consequence of publications in LUBM having no incoming edges. Likewise, it is sensible that the MAGNN has learned that research assistants are graduate students, but it has also erroneously learned that all graduate students are research assistants. Finally, note that the fourth rule in the table is subsumed by the third rule — this is because they come from two different MAGNNs, which have learned different sound rules. Table 10 of Appendix C provides a longer list of explanatory rules that have been reduced in size.

## 5 Discussion

**Our Contribution**    We considered mean-GNNs with non-negative weights (MAGNNs), showing first that they can express logical functions that go beyond FOL. We proved which ELUQ rules (a class of monotonic ALCQ rules) can be sound for MAGNNs, which turned out to be very limited. The resulting rule space is finite, which enabled us to define a procedure to check for all sound ELUQ rules. We also provided a restricted fragment of FOL, from which sound rules can be constructed to explain any prediction of a MAGNN. In our experiments, we found that mean-GNNs still perform well on benchmark datasets when restricted to having non-negative weights. We also found a variety of sound monotonic rules on half the datasets, with provably almost no sound monotonic rules on the other half. This, alongside several nonsensical sound rules, raise questions about how mean-GNNs can be encouraged to recover the underlying rules in the datasets, and suggests that max- or sum-GNNs may be preferable when provable soundness is required. It also shows the importance of rule extraction techniques, given the good empirical performance of MAGNNs on the test datasets but nonsensical rules they have learned (for example, that everything is a publication). Finally, we computed explanatory rules for predictions made by MAGNNs on the LUBM dataset.

**Logical Explanations for GNNs**    Other works also extract logical explanations for GNN predictions [4, 5, 20, 28]. However, none of these provide theoretical guarantees that the explanations are sound for the model, only empirical evidence that they are approximately faithful. Pluska et al. [28], for example, learn decision trees from GNNs and then extract logical rules from the decision trees — they are not equivalent to the GNNs they are derived from, so the extracted logical rules are not provably sound. Furthermore, Köhler and Heindorf [20] only consider explanations in EL, which is a fragment of ELUQ.

**Mean-GNN Theory**    Further works also provide theoretical analyses of GNNs with mean aggregation. Vasileiou et al. [39] provide a generalisation bound for mean-GNNs. Adam-Day et al. [2] show that mean-GNNs used for graph classification tend to a constant function as the sizes of the input graphs tend to infinity (under certain random graph models). Their approach of tending the size to infinity is similar to our technique used in the proof of Theorem 3. In contemporaneous work, Schönherr and Lutz [33] identify the FOL expressivity of mean-GNNs in the uniform (standard) and non-uniform (only on graphs up to a bounded number of nodes) settings. In the uniform setting, they prove that any continuous mean-GNN (that is equivalent to a FOL classifier) has an equivalent logical classifier expressible in alternation-free modal logic (AFML). However, they do not provide a construction of the AFML classifier, a demonstration that such a construction would be obtainable in practice, or a means to recover sound rules if the mean-GNN is not equivalent to a FOL classifier.

**Limitations**    Our work is limited in several ways. First, despite being able to check the infinite space of ELUQ rules for soundness by instead checking finitely many rules of the form given in Theorem 3, the size of this finite space grows exponentially in the number of predicates, making the search intractable as the number of predicates increases (as seen on FB237v1, for example). Also, it is an open question as to whether ELUQ is a maximal monotonic fragment of ALCQ, so there may be other monotonic rules that can be sound for mean-GNNs. Finally, we only consider the $\leq$ operator for the mean-GNN step classification function $\texttt{cls}_t$ due to its standard use in prior work [25, 35, 37] and the fact that using $<$ yields an entirely different set of theoretical results (for example, rules with a body of $\exists P.A$ could be sound for such a mean-GNN, without $\exists P.\top$ being sound).

## Acknowledgments and Disclosure of Funding

Matthew Morris is funded by an EPSRC scholarship (CS2122_EPSRC_1339049). This work was also supported by Samsung Research UK, the EPSRC projects UKFIRES (EP/S019111/1) and ConCur (EP/V050869/1). The authors would like to acknowledge the use of the University of Oxford Advanced Research Computing (ARC) facility in carrying out this work: http://dx.doi.org/10.5281/zenodo.22558.

For the purpose of Open Access, the authors have applied a CC BY public copyright licence to any Author Accepted Manuscript (AAM) version arising from this submission.

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

# A  Proofs

## A.1  Proposition 1

**Proposition.** *There exists a MAGNN $\mathcal{M}$ such that for any dataset $D$ and constant $a \in \mathsf{con}(D)$, $U(a) \in T_{\mathcal{M}}(D)$ if and only at least half of the neighbours $b$ of $a$ in $D$ are such that $U(b) \in D$. This logical function cannot be defined in FOL [21].*

*Proof.* Consider a MAGNN $\mathcal{M}$ over a signature with only one unary predicate ($U$) and only one binary predicate ($P$), corresponding to colour $c$ in the canonical encoding. Let $\mathcal{M}$ have $L = 1$ (one layer), with $\mathbf{A}_1 = (0)$, $\mathbf{b}_1 = (0)$, and $\mathbf{B}_1^c = (1)$. Finally, let $\sigma_1$ be the identity function and $t_{\mathcal{M}} = 0.5$.

Consider a dataset $D$ and constant $a$ in $D$. Then $\mathbf{v}_L^a$, the labelling of $v^a$ after $\mathcal{M}$ is applied to $\mathsf{enc}(D)$, depends only on the number and type of the direct neighbours of $v^a$. Each neighbour $v^b$ either has $U(b) \in D$ or $U(b) \notin D$. We end up with

$$\mathbf{v}_L^a[1] = \frac{|\{v^b \mid (v^a, v^b) \in E^c \text{ and } \mathbf{v}_L^b[1] = 1\}|}{|N_c(v^a)|}$$
$$= \frac{|\{b \mid P(a,b) \in D \text{ and } U(b) \in D\}|}{|\{b \mid P(a,b) \in D\}|}$$

Now $\mathbf{v}_L^a[1] \geq 0.5 = t_{\mathcal{M}}$ if and only if $|\{b \mid P(a,b) \in D \text{ and } U(b) \in D\}| \geq |\{b \mid P(a,b) \in D \text{ and } U(b) \notin D\}|$. I.e. $U(a) \in T_{\mathcal{M}}(D)$ if and only at least half of the neighbours $b$ of $a$ in $D$ are such that $U(b) \in D$.

However, this logical function cannot be expressed in FOL. To prove this, we consider Libkin [21], who defines the majority predicate $\mathrm{MAJ}(\varphi, \psi)$, which tests if all constants satisfying $\varphi$ contains at least half of the constants satisfying $\psi$. We find that, for every dataset $D$ and constant $a \in \mathsf{con}(D)$, we have $(D, a) \models \mathrm{MAJ}(R(a, x) \wedge U(x), R(a, x))$ if and only if "at least half of the neighbours $b$ of $a$ in $D$ are such that $U(b) \in D$" if and only if $U(a) \in T_{\mathcal{M}}(D)$.

However, Libkin [21, Corollary 13.17] proves that MAJ queries cannot be defined in an extension of FOL, from which it trivially follows that they cannot be defined in FOL. Thus, $\mathcal{M}$ is equivalent to a logical function that cannot be defined in FOL. $\qquad\square$

## A.2  Theorem 3

**Theorem.** *Let $\mathcal{M}$ be a MAGNN and $r$ an ELUQ rule that is sound for $\mathcal{M}$. Then there exists a finite set of rules $R$ such that:*

1. *Each $r' \in R$ has the form $\exists P_1.\top \sqcap ... \sqcap \exists P_j.\top \sqcap A_1 \sqcap ... \sqcap A_k \sqcap \top \sqsubseteq A_{k+1}$, where each $P_i$ is a binary predicate, $A_i$ a unary predicate, and $j, k \in \mathbb{N}_0$.*

2. *Each $r' \in R$ is sound for $\mathcal{M}$.*

3. *$R$ subsumes $r$.*

*Proof.* First, notice that $r$ is of the form $C^1 \sqcup ... \sqcup C^m \sqsubseteq A$ for ELUQ concepts $C^1, ..., C^m$ containing no disjunction in the outer scope (i.e. disjunction is only nested within existential quantification), $m \in \mathbb{N}$, and atomic concept $A$. Note that $r$ not containing any disjunctions in the outer scope is covered by $m = 1$.

Now, let $R' := \{C^1 \sqsubseteq A, ..., C^m \sqsubseteq A\}$. By the operation of rule application, $T_{R'}(D) = T_r(D)$ for all datasets $D$. Also, each rule in $R'$ is sound for $\mathcal{M}$, as a consequence of $r$ being sound for $\mathcal{M}$. We will construct $R$ by including in it a rule for each rule in $R'$.

**Base Case**

Consider each $r_1 \in R'$. If $r_1$ contains no $\exists$ or $\geq_n$ operators, then it is already trivially of the form specified in the theorem, so let $r_k = r_1$ and include it in $R$. Also, as stated above, $r_1$ is sound for $\mathcal{M}$. Otherwise, we have that $r_1$ is of the form $\geq_n P.C_1 \sqcap C_2 \sqsubseteq A$, where $P$ is a binary predicate, $C_1$ an ELUQ concept, $C_2$ an ELUQ concept without disjunction, and $n$ a positive integer.

We now inductively define a sequence $r_2, ..., r_k$ of ELUQ rules that such for all $i \in \{1, ..., k-1\}$,

1. $r_{i+1}$ is sound for $\mathcal{M}$

2. $r_{i+1}$ subsumes $r_i$

### Inductive Step

For this purpose, consider the last defined rule $r_i$: we define $r_{i+1}$ and prove the above properties.

If $r_i$ contains no $\exists$ or $\geq_n$ operators, then let $k := i$ and cease the inductive construction. As assumed in the inductive hypothesis, $r_i$ is sound for $\mathcal{M}$. Otherwise, we have that $r_i$ is of the form $\geq_n P.C_1 \sqcap C_2 \sqsubseteq A$, where $P$ is a binary predicate, $C_1$ an ELUQ concept, $C_2$ an ELUQ concept without disjunction, and $n$ a positive integer.

Now let $r_{i+1}$ be the rule $\exists P.\top \sqcap C_2 \sqsubseteq A$. Trivially, $r_{i+1}$ subsumes $r_i$, since they have the same head and whenever the body of $r_{i+1}$ is satisfied, so too is the body of $r_i$. It remains to prove that $r_{i+1}$ is sound for $\mathcal{M}$. For this purpose, assume to the contrary that $r_{i+1}$ is not sound for $\mathcal{M}$.

If for all datasets $D_1$ such that $P(a,b) \in D_1$ and $(D_1, a) \models C_2$ for some constant $b$, we have $A(a) \in T_{\mathcal{M}}(D_1)$, then $r_{i+1}$ would be sound for $\mathcal{M}$. This follows from these conditions covering all datasets that satisfy the body of $r_{i+1}$.

Likewise, if for all datasets $D_1$ such that $P(a,b) \in D_1$, $(D_1, a) \models C_2$, $b \neq a$, and $b$ is not mentioned in any other fact of $D_1$, we have $A(a) \in T_{\mathcal{M}}(D_1)$, then $r_{i+1}$ would be sound for $\mathcal{M}$. This is because neither mentioning $b$ in another fact of $D_1$ nor having $a = b$ will decrease the output of $\mathcal{M}(D_1)$ for node $v^a$,

So instead there exists some dataset $D_1$ such that $P(a,b) \in D_1$, $(D_1, a) \models C_2$, and $A(a) \notin T_{\mathcal{M}}(D_1)$ for some constants $a \neq b$, where $b$ is not mentioned in any fact of $D_1$ besides $P(a,b)$. Here follows a representation of $D_1$:

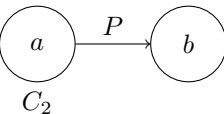

Now consider $\mathbf{v}_L^a$, the output for node $v^a$ when $\mathcal{M}$ is applied to $\mathrm{enc}(D_1)$, and let $1 \leq p \leq \delta$ be in the index of the unary predicate $A$ in the canonical encoding. Since $A(a) \notin T_{\mathcal{M}}(D_1)$, we must have $\mathbf{v}_L^a[p] < t_{\mathcal{M}}$. Let $t := \frac{t_{\mathcal{M}} - \mathbf{v}_L^a[p]}{2}$.

We now define a dataset $D_m$, parameterized by a non-negative integer $m$. We define $D_m = D_1 \setminus \{P(a,b)\} \cup \{P(a, c_1), ..., P(a, c_n)\} \cup \{P(a, b_1), ..., P(a, b_m)\}$ for distinct constants $c_1, ..., c_n, b_1, ..., b_m$ (all also distinct from $a$) and further require that $(D_m, c_1) \models C_1, ..., (D_m, c_n) \models C_1$, by including other facts necessary for the satisfaction to hold. We also require that $b_1, ..., b_m$ are not mentioned in any other facts of $D_m$. These further requirements may be satisfied in several different ways, but any of them will suffice for the definition of $D_m$. Here follows a representation of $D_m$:

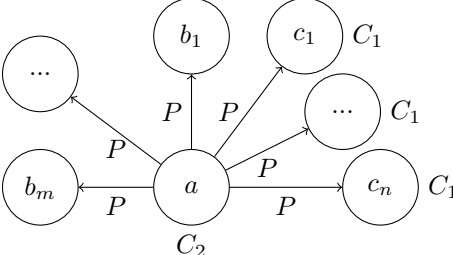

Notice that $(D_m, a) \models C_2$ and $(D_m, a) \models \geq_n P.C_1$, so $A(a) \in T_{r_i}(D_m)$. But from the soundness of $r_i$, we also have $A(a) \in T_{\mathcal{M}}(D_m)$. However, as $m \to \infty$, the computation in node $v^a$ when $\mathcal{M}$ is applied to $\text{enc}(D_m)$ tends to that of $\mathcal{M}$ being applied to $\text{enc}(D_1)$ — this is consequence of the definition of MAGNNs. Thus, there exists some $m$ such that when considering $\mathbf{v}_L^a$, the output of $\mathcal{M}$ applied to $\text{enc}(D_m)$, we have $\mathbf{v}_L^a[p] < t$ (note that this requires the use of continuous activation functions, otherwise it is possible that $\mathbf{v}_L^a[p] \geq t$ for every $m$).

But since $t < t_{\mathcal{M}}$, we have $\mathbf{v}_L^a[p] < t_{\mathcal{M}}$. And since $A(a) \in T_{\mathcal{M}}(D_m)$, we have $\mathbf{v}_L^a[p] \geq t_{\mathcal{M}}$. This is a contradiction, so $r_{i+1}$ is sound for $\mathcal{M}$.

#### End of Inductive Construction

Since $r_1$ contains a finite number of $\exists$ or $\geq_n$ operators and each $r_{i+1}$ has one fewer $\exists$ or $\geq_n$ operator than $r_i$, this inductive construction is guaranteed to terminate for some $k$.

We thus have a sequence $r_1, r_2, ..., r_k$ of ELUQ rules such that for all $i \in \{1, ..., k-1\}$, $r_{i+1}$ subsumes $r_i$. Since $r_{i+1}$ subsumes $r_i$, for all datasets $D$ we have $T_{r_i}(D) \subseteq T_{r_{i+1}}(D)$. So $T_{r_1}(D) \subseteq T_{r_2}(D) \subseteq ... \subseteq T_{r_k}(D)$, and thus $T_{r_1}(D) \subseteq T_{r_k}(D)$. Also, notice that $r_k$ contains no existential quantifiers or disjunction, so it is of the form specified in the theorem. Furthermore, $r_k$ is sound for $\mathcal{M}$, as was shown above. We include $r_k$ in $R$.

#### Conclusion

We now have a set $R$ of rules of the form specified in the theorem, such that each is sound for $\mathcal{M}$. It remains to be shown that $R$ subsumes $r$.

Let $D$ be a dataset. Recall that $T_r(D) = T_{R'}(D)$. But for each $r_1 \in R'$, there exists $r_k \in R$ such that $T_{r_1}(D) \subseteq T_{r_k}(D)$. Thus $\bigcup_{r_1 \in R'} T_{r_1}(D) \subseteq \bigcup_{r_k \in R} T_{r_k}(D)$, and so $T_{R'}(D) \subseteq T_R(D)$. Hence, we have $T_r(D) \subseteq T_R(D)$, so $R$ subsumes $r$, as required.

$\square$

### A.3 Proposition 4

**Proposition.** *Let $\mathcal{M}$ be a MAGNN and $r : \exists P_1.\top \sqcap ... \sqcap \exists P_j.\top \sqcap A_1 \sqcap ... \sqcap A_k \sqsubseteq A_{k+1}$ be a rule of the form given in Theorem 3. Define $D_{base} := \{A_1(a), ..., A_k(a), P_1(a, b), ..., P_j(a, b)\}$ for constants $a \neq b$. Then $r$ is sound for $\mathcal{M}$ if and only if $A_{k+1}(a) \in T_{\mathcal{M}}(D_{base})$.*

*Proof.* We prove both direction of the if and only if. First, assume $r$ is sound for $\mathcal{M}$. Notice that $(D_{\text{base}}, a) \models B$, where $B$ denotes the body concepts of $r$, so we have $A_{k+1}(a) \in T_r(D_{\text{base}})$. But since $r$ is sound for $\mathcal{M}$, $T_r(D_{\text{base}}) \subseteq T_{\mathcal{M}}(D_{\text{base}})$ and thus $A_{k+1}(a) \in T_{\mathcal{M}}(D_{\text{base}})$ as required.

Now assume that $A_{k+1}(a) \in T_{\mathcal{M}}(D_{\text{base}})$. We prove that $r$ is sound for $\mathcal{M}$. For this purpose, let $D$ be an arbitrary dataset: we prove that $T_r(D) \subseteq T_{\mathcal{M}}(D)$. So let $A_{k+1}(c) \in T_r(D)$.

#### Extending The Base Dataset

Consider the set $D_{\text{splits}}$, consisting of every dataset $\{A_1(a), ..., A_k(a), P_1(a, b_1), ..., P_j(a, b_j)\}$ for (not necessarily mutually distinct) constants $b_1, ..., b_j$ that are distinct to $a$. Then for each such $D_1 \in D_{\text{splits}}$, the computation of $\mathcal{M}(D_1)$ is identical to that of $\mathcal{M}(D_{\text{base}})$. So $A_{k+1}(a) \in T_{\mathcal{M}}(D_1)$.

Now consider the set $D_{\text{merges}}$, consisting of every dataset $\{A_1(a), ..., A_k(a), P_1(a, b_1), ..., P_j(a, b_j)\}$ for (not necessarily mutually distinct) constants $b_1, ..., b_j$ that are not necessarily distinct to $a$. Each $D_2 \in D_{\text{merges}}$ either has $a$ distinct to all $b_1, ..., b_j$, or it does not. If it does, then $D_2 \in D_{\text{splits}}$, in which case $A_{k+1}(a) \in T_{\mathcal{M}}(D_2)$ Otherwise, $D_2$ has a corresponding $D_1 \in D_{\text{splits}}$ where each $b_i$ such that $b_i = a$ is swapped out for another constant $b_i' \neq a$.

Consider $\mathbf{v}_L^a$ and $\mathbf{v}_L^{a\,\prime}$, the outputs for node $v^a$ when $\mathcal{M}$ is applied to $D_1$ and $D_2$ respectively. Notice that, by the operations of MAGNNs, for all $p \in \{1, ..., \delta\}$ we have $\mathbf{v}_L^a[p] \leq \mathbf{v}_L^{a\,\prime}[p]$. This is because merging each of the neighbours $b_i'$ of $a$ into $b_i = a$ will never decrease the output of the MAGNN.

Now let $p$ be the index of $A_{k+1}$ in the canonical encoding. Then since $\mathbf{v}_L^a[p] \leq \mathbf{v}_L^a{}'[p]$ and $A_{k+1}(a) \in T_{\mathcal{M}}(D_1)$, we have $t_{\mathcal{M}} \leq \mathbf{v}_L^a[p] \leq \mathbf{v}_L^a{}'[p]$, and thus $A_{k+1}(a) \in T_{\mathcal{M}}(D_2)$. So for every $D_2 \in D_{\text{merges}}$, we have $A_{k+1}(a) \in T_{\mathcal{M}}(D_2)$.

## Conclusion

We now prove that $A_{k+1}(c) \in T_{\mathcal{M}}(D)$. Since $A_{k+1}(c) \in T_r(D)$, there exists (not necessarily distinct) constants $d_1, ..., d_j$ in $D$ such that $D' := \{A_1(c), ..., A_k(c), P_1(c, d_1), ..., P_j(c, d_j)\} \subseteq D$. There are potentially multiple such constants $d_i$ that could be used for $D' \subseteq D$. For each $i \in \{1, ..., j\}$, if there exists constant $d_i \neq a$ such that $P_i(c, d_i) \in D$, then choose this $d_i$ to be in $D'$.

But then, up to the relabelling of the constant $c \mapsto a$, we have $D' \in D_{\text{merges}}$, so $A_{k+1}(c) \in T_{\mathcal{M}}(D')$, since MAGNNs are agnostic to the specific constants used.

$D$ is a superset of $D'$. Consider $v^c \in \text{enc}(D')$ and the application of $\mathcal{M}$ to $D'$ and $D$, resulting in $\mathbf{v}_L^c{}'$ and $\mathbf{v}_L^c$, respectively. Any unary facts in $D$ that are not in $D'$ will preserve $\mathbf{v}_L^c{}' \leq \mathbf{v}_L^c$, from the definition of MAGNNs and the canonical encoding. The only binary facts in $D$ that are not in $D'$ that could decrease the computation of $\mathbf{v}_L^c{}'$ (leading to $\mathbf{v}_L^c{}'[p] > \mathbf{v}_L^c[q]$ for some index $q$) are of the form $P_i(c, d_i)$ for $c \neq d_i$, where $D'$ contains some fact $P_i(c, c)$. However, as was stated in the definition of $D'$, if $D$ contains a fact $P_i(c, d_i)$ with $c \neq d_i$, then $D'$ will not contain $P_i(c, c)$. So, no facts added when extending $D'$ to $D$ will decrease the computation of $\mathbf{v}_L^c{}'$, and we obtain $\mathbf{v}_L^c{}' \leq \mathbf{v}_L^c$.

Now let $p$ be the index of $A_{k+1}$ in the canonical encoding. Then since $A_{k+1}(c) \in T_{\mathcal{M}}(D')$, we have $t_{\mathcal{M}} \leq \mathbf{v}_L^c{}'[p] \leq \mathbf{v}_L^c[p]$, and thus $A_{k+1}(c) \in T_{\mathcal{M}}(D)$, as required.

$\square$

## A.4 Lemma 7

We provide a lemma that is used in the proof of Theorem 6.

**Lemma 7.** *Let $\mathcal{M}$ be a MAGNN, $D_a$ a dataset, and $a \in \text{con}(D_a)$ a constant. For any $\ell \in \mathbb{N}_0$ and $c \in \text{con}(D_a)$, we have $(D_a, c) \models C_\ell^c$, where $C_\ell^c$ is dependent on $D_a$.*

*Proof.* We prove this by induction on $\ell$.

## Base Case

For $\ell = 0$ and some $c \in \text{con}(D_a)$, we have $C_0^c = \top \sqcap A_1 \sqcap ... \sqcap A_k$, where $A_1, ..., A_k$ are all atomic concepts $A_i$ such that $A_i(c) \in D_a$. But then $(D_a, c) \models A_i$ for every $A_i$ in the conjunction of $C_0^c$, so $(D_a, c) \models C_0^c$.

## Inductive Step

Consider some $\ell \in \mathbb{N}$, and assume that the claim holds for $\ell - 1$. Let $c \in \text{con}(D_a)$. We have $C_\ell^c = A_1 \sqcap ... \sqcap A_k \sqcap C'$, where $A_1, ..., A_k$ are all atomic concepts $A_i$ such that $A_i(c) \in D_a$ and $C'$ denotes the remainder of the concept. Similarly to the base case, we obtain $(D_a, c) \models A_1 \sqcap ... \sqcap A_k$.

$C'$ is a conjunction of $\top$ with a concept $C_P$ for each binary predicate $P$ such that there exists $P(c, c_i) \in D_a$. So consider such a $P$: we prove that $(D_a, c) \models C_P$.

Let $c_1, ..., c_n$ be all the constants $c_i \in \text{con}(D_a)$ such that $P(c, c_i) \in D$. Then we have $C_P := \exists_n P.(C_{\ell-1}^{c_1}, ..., C_{\ell-1}^{c_n}) \sqcap \leq_n P.\top$. Since $c$ has exactly $n$ $P$-neighbours in $D_a$, we have $(D_a, c) \models \leq_n P.\top$. By induction, $(D_a, c_i) \models C_{\ell-1}^{c_i}$ for every $i \in \{1, ..., n\}$. Also, all $c_1, ..., c_n$ are distinct. So from the semantics of $\exists_n$, $(D_a, c) \models \exists_n P.(C_{\ell-1}^{c_1}, ..., C_{\ell-1}^{c_n})$. Thus, we obtain $(D_a, c) \models C_P$.

Now since $(D_a, c) \models C_P$ holds for every binary predicate $P$, we have $(D_a, c) \models C'$, and thus $(D_a, c) \models C_\ell^c$, as required.

$\square$

## A.5 Theorem 6

**Theorem.** *For any MAGNN $\mathcal{M}$, dataset $D_a$, and fact $A(a) \in T_{\mathcal{M}}(D_a)$, the rule $r : C_L^a \sqsubseteq A$ (dependent on $D_a$) is sound for $\mathcal{M}$, and $A(a) \in T_r(D_a)$.*

*Proof.* First, we have $A(a) \in T_r(D_a)$, since $A$ is the head of $r$ and $(D_a, a) \models C_L^a$ follows directly from Lemma 7. It remains to be shown that $r$ is sound for $\mathcal{M}$. Let $D_b$ be a dataset: to show soundness, we prove that $T_r(D_b) \subseteq T_{\mathcal{M}}(D_b)$. For this purpose, let $A(b) \in T_r(D_b)$ be an arbitrary fact; we prove $A(b) \in T_{\mathcal{M}}(D_b)$.

### Minimal Satisfaction Dataset

We define a dataset $D_L^a$ in an analogous fashion to the definition of $C_L^a$, to be the $L$-hop neighbourhood of the constant $a$ in $D_a$.

Given $c \in \mathrm{con}(D_a)$, $\ell \in \mathbb{N}_0$, we inductively define a dataset $D_\ell^c$ by $D_\ell^c := \{A_i(c) \mid A_i(c) \in D_a\}$. Then, if $\ell = 0$, we are done: $D_\ell^c$ has been defined. Otherwise, for each binary predicate $P$, let $c_1, ..., c_n$ be all the constants $c_i \in \mathrm{con}(D_a)$ such that $P(c, c_i) \in D_a$. Then, if $n > 0$, extend $D_\ell^c$ as follows: $D_\ell^c := D_\ell^c \cup D_{\ell-1}^{c_1} \cup D_{\ell-1}^{c_2} \cup ... \cup D_{\ell-1}^{c_n} \cup \{P(c, c_i) \mid P(c, c_i) \in D_a\}$. $D_\ell^c$ is extended for each binary predicate $P$. Note that each $D_{\ell-1}^{c_i}$ is defined inductively.

At this point in the proof, we have three datasets to consider:

- $D_a$ — the original input dataset from which a fact $A(a)$ was produced by the MAGNN $T_{\mathcal{M}}$

- $D_b$ — an arbitrary dataset instantiated for the proof of soundness

- $D_L^a$ — the $L$-hop neighbourhood of the constant $a$ in $D_a$

In the remainder of the proof, we show the following in sequence:

(1) $A(a) \in T_{\mathcal{M}}(D_L^a)$, since $A(a) \in T_{\mathcal{M}}(D_a)$ — follows from definition of $D_L^a$ as the $L$-hop neighbourhood.

(2) $A(b) \in T_{\mathcal{M}}(D_b)$, since $A(a) \in T_{\mathcal{M}}(D_L^a)$ — follows from the fact that any differences between $D_b$ and $D_L^a$ cannot decrease the MAGNN output for $v^a / v^b$, plus a mapping between constants of $D_b$ and $D_L^a$.

### (1) Prove $A(a) \in T_{\mathcal{M}}(D_L^a)$ Using $A(a) \in T_{\mathcal{M}}(D_a)$

First note that $D_L^a \subseteq D_a$, since only facts from $D_a$ are included in its definition. Consider $\mathbf{v}_L^a, \mathbf{v}_L^{a\,\prime}$, the output of $\mathcal{M}$ on $D_a, D_L^a$ respectively for the vertex corresponding to $a$, and let $p$ be the index of $A$ in the canonical encoding. Since $A(a) \in T_{\mathcal{M}}(D_a)$, $\mathbf{v}_L^a[p] \geq t_{\mathcal{M}}$. However, since $\mathcal{M}$ has only $L$ layers, the computation of $\mathbf{v}_L^a$ is affected only by vertices in the $L$-hop neighbourhood of $v^a$.

Since $D_L^a$ contains exactly the $L$-hop neighbourhood of $a$, we thus have $\mathbf{v}_L^a = \mathbf{v}_L^{a\,\prime}$. This implies $\mathbf{v}_L^{a\,\prime}[p] \geq t_{\mathcal{M}}$ and thus $A(a) \in T_{\mathcal{M}}(D_L^a)$.

### (2) Prove $A(b) \in T_{\mathcal{M}}(D_b)$ Using $A(a) \in T_{\mathcal{M}}(D_L^a)$

First, we inductively construct a mapping $\pi : \mathrm{con}(D_L^a) \to \mathrm{con}(D_b)$ from constants to constants. Intuitively, this will track "corresponding" constants from the two datasets in the computation of $\mathcal{M}$. We also assign a "level" to each constant of $\mathrm{con}(D_L^a)$ from the set $\{0, ..., L\}$. We perform the induction on $\ell$ from $L$ to $0$, simultaneously proving that for each constant $c \in \mathrm{con}(D_L^a)$ of level $\ell$, we have:

$$(D_L^a, c) \models C_\ell^c \text{ and } (D_b, \pi(c)) \models C_\ell^c \tag{3}$$

### Inductive Construction of Constant Mapping

For the base case, we define $\pi(a) = b$, and let the level of $a$ be $L$. Since $A(b) \in T_r(D_b)$, $(D_b, b) \models C_L^a$, the body of $r$. So we have $(D_b, \pi(a)) \models C_L^a$ and also trivially $(D_L^a, a) \models C_L^a$.

Now consider each constant $c \in \text{con}(D_L^a)$ of level $\ell \geq 1$; we define $\pi$ for constants of level $\ell - 1$. For each binary predicate $P$, let $c_1, ..., c_n$ be all the constants $c_i \in \text{con}(D_a)$ such that $P(c, c_i) \in D_a$. Let the level of each $c_1, ..., c_n$ be $\ell - 1$.

Let $d := \pi(c)$. By the inductive assumption, $(D_b, \pi(c)) \models C_\ell^c$, so we have that $(D_b, d) \models \exists_n P.(C_{\ell-1}^{c_1}, ..., C_{\ell-1}^{c_n}) \sqcap \leq_n P.\top$. So there exist distinct constants $d_1, ..., d_n \in \text{con}(D_b)$ such that $(D_b, d_1) \models C_{\ell-1}^{c_1}, ..., (D_b, d_n) \models C_{\ell-1}^{c_n}$ and $P(d, d_1) \in D_b, ..., P(d, d_n) \in D_b$. We define $\pi(c_1) = d_1, ..., \pi(c_n) = d_n$. Notice that we have that $(D_b, \pi(c_1)) \models C_{\ell-1}^{c_1}, ..., (D_b, \pi(c_n)) \models C_{\ell-1}^{c_n}$, and trivially from the construction of $D_L^a$ that $(D_L^a, c_1) \models C_{\ell-1}^{c_1}, ..., (D_b, c_n) \models C_{\ell-1}^{c_n}$, as required for the induction proof.

### Monotonicity

For each constant $c \in \text{con}(D_L^a)$ and $d \in \text{con}(D_b)$, let $\mathbf{v}_L^c$, $\bar{\mathbf{v}}_L^d$ be the outputs when $\mathcal{M}$ is applied to $D_L^a$ and $D_b$, respectively. Likewise, let $E^o$, $\bar{E}^o$ denote the edges and $N_o$, $\bar{N}_o$ the neighbours in $\text{enc}(D_L^a)$ and $\text{enc}(D_b)$ for colour $o$, respectively. We prove the following claim by induction on $\ell$ from 0 to $L$.

**Claim:** For each $\ell \in \{0, ..., L\}$, constant $c \in \text{con}(D_L^a)$ of level $\ell' \in \{\ell, ..., L\}$, and $i \in \{1, ..., \delta_\ell\}$, we have $\mathbf{v}_\ell^c[i] \leq \bar{\mathbf{v}}_\ell^{\pi(c)}[i]$.

### Base Case

For the base case ($\ell = 0$), let $i \in \{1, ..., \delta_0\}$, let $\ell' \in \{0, ..., L\}$, and consider each constant $c \in \text{con}(D_L^a)$ of level $\ell'$.

As proven in Equation (3), we have $(D_L^a, c) \models C_{\ell'}^c$ and $(D_b, \pi(c)) \models C_{\ell'}^c$. Recall that $C_{\ell'}^c$ includes $\top \sqcap A_1 \sqcap ... \sqcap A_k$, where $A_1, ..., A_k$ are all atomic concepts $A'$ such that $A'(c) \in D_a$. From the definition of $D_L^a$, the unary facts in $D_L^a$ that mention $c$ are precisely $A_1(c), ..., A_k(c)$. Then since $(D_b, \pi(c)) \models C_{\ell'}^c$, we have $A_1(\pi(c)), ..., A_k(\pi(c)) \in D_b$.

So if $\mathbf{v}_0^c[i] = 1$, then $A_i(c) \in D_L^a$, where $A_i$ is the unary predicate with index $i$ in the canonical encoding. This implies that $A_i(\pi(c)) \in D_b$, and thus $\bar{\mathbf{v}}_0^{\pi(c)}[i] = 1$. This proves the base case, since $\mathbf{v}_0^c[i] = 1 \implies \bar{\mathbf{v}}_0^{\pi(c)}[i] = 1$ and the definition of the canonical encoding let us conclude that $\mathbf{v}_0^c[i] \leq \bar{\mathbf{v}}_0^{\pi(c)}[i]$.

### Inductive Step

For the inductive step, we prove the claim for $\ell \geq 1$; to do so, assume the claim holds for $\ell - 1$. Let $\ell' \in \{\ell, ..., L\}$, let $i \in \{1, ..., \delta_\ell\}$, and consider each constant $c \in \text{con}(D_L^a)$ of level $\ell'$. We prove that $\mathbf{v}_\ell^c[i] \leq \bar{\mathbf{v}}_\ell^{\pi(c)}[i]$. Consider the computation of $\mathbf{v}_\ell^c[i]$ and $\bar{\mathbf{v}}_\ell^{\pi(c)}[i]$:

$$\mathbf{v}_\ell^c[i] = \sigma_\ell \left( \mathbf{b}_\ell[i] + \sum_{j=1}^{\delta_{\ell-1}} \mathbf{A}_\ell[i,j]\mathbf{v}_{\ell-1}^c[j] + \sum_{o \in \text{Col}} \sum_{j=1}^{\delta_{\ell-1}} \mathbf{B}_\ell[i,j] \, \text{mean}(\{\!\{ \mathbf{u}_{\ell-1}[j] \mid (v^c, u) \in E^o \}\!\}) \right)$$

$$\tag{4}$$

$$\bar{\mathbf{v}}_\ell^{\pi(c)}[i] = \sigma_\ell \left( \mathbf{b}_\ell[i] + \sum_{j=1}^{\delta_{\ell-1}} \mathbf{A}_\ell[i,j]\bar{\mathbf{v}}_{\ell-1}^{\pi(c)}[j] + \sum_{o \in \text{Col}} \sum_{j=1}^{\delta_{\ell-1}} \mathbf{B}_\ell[i,j] \, \text{mean}(\{\!\{ \bar{\mathbf{u}}_{\ell-1}[j] \mid (v^{\pi(c)}, u) \in \bar{E}^o \}\!\}) \right)$$

$$\tag{5}$$

From the induction hypothesis, since $c$ is of level $\ell' \geq \ell - 1$, we have $\mathbf{v}^c_{\ell-1}[j] \leq \bar{\mathbf{v}}^{\pi(c)}_{\ell-1}[j]$. Since all else in the above equations is equal, it remains to be shown that, for all colours $o \in \mathrm{Col}$ and $j \in \{1, ..., \delta_{\ell-1}\}$:

$$\mathrm{mean}(\{\!\{\mathbf{u}_{\ell-1}[j] \mid (v^c, u) \in E^o\}\!\}) \leq \mathrm{mean}(\{\!\{\bar{\mathbf{u}}_{\ell-1}[j] \mid (v^{\pi(c)}, u) \in \bar{E}^o\}\!\})$$

So let $o \in \mathrm{Col}$, $j \in \{1, ..., \delta_{\ell-1}\}$, and let $P$ be the binary predict corresponding to colour $o$. Now if there exists no $c_k \in \mathrm{con}(D_a)$ such that $P(c, c_k) \in D_a$, then by the definition of $E^o$ as the edges from $\mathrm{enc}(D^a_L)$, we have $E^o = \emptyset$ and thus $\mathrm{mean}(\{\!\{\mathbf{u}_{\ell-1}[j] \mid (v^c, u) \in E^o\}\!\}) = 0$. So the inequality trivially holds, since each $\bar{\mathbf{u}}_{\ell-1}[j]$ is non-negative.

Thus, instead consider all $c_1, ..., c_n \in \mathrm{con}(D_a)$ such that $P(c, c_k) \in D_a$. Note that each such $c_k$ has level $\ell' - 1$. Then since $\ell' - 1 \geq \ell - 1$, from the inductive hypothesis, we have $\mathbf{v}^{c_k}_{\ell-1}[j] \leq \bar{\mathbf{v}}^{\pi(c_k)}_{\ell-1}[j]$ for each $c_k$.

Furthermore, since $c \in \mathrm{con}(D^a_L)$ has level $\ell'$, we obtain $(D_b, \pi(c)) \models C^c_{\ell'}$ from Equation (3). Thus, we have $(D_b, \pi(c)) \models \exists_n P.(C^{c_1}_{\ell'-1}, ..., C^{c_n}_{\ell'-1}) \sqcap \leq_n P.\top$. So there exist exactly $n$ distinct constants $d_1, ..., d_n \in \mathrm{con}(D_b)$ such that $P(\pi(c), d_1) \in D_b, ..., P(\pi(c), d_n) \in D_b$. However, recall from the definition of $\pi$, that these constants are precisely $d_1 = \pi(c_1), ..., d_n = \pi(c_n)$.

From the canonical encoding, we thus have a one-to-one correspondence between the sets $\{v^{c_k} \mid (v^c, v^{c_k}) \in E^o\}$ and $\{v^{\pi(c_k)} \mid (v^{\pi(c)}, v^{\pi(c_k)}) \in \bar{E}^o\} = \{v^{d_k} \mid (v^{\pi(c)}, v^{d_k}) \in \bar{E}^o\}$. Also, as stated above, from the inductive hypothesis, we have $\mathbf{v}^{c_k}_{\ell-1}[j] \leq \bar{\mathbf{v}}^{\pi(c_k)}_{\ell-1}[j]$ for each $c_k$. Combining these, we obtain the inequality $\mathrm{mean}(\{\!\{\mathbf{u}_{\ell-1}[j] \mid (v^c, u) \in E^o\}\!\}) \leq \mathrm{mean}(\{\!\{\bar{\mathbf{u}}_{\ell-1}[j] \mid (v^{\pi(c)}, u) \in \bar{E}^o\}\!\})$. This allows us to conclude $\mathbf{v}^c_\ell[i] \leq \bar{\mathbf{v}}^{\pi(c)}_\ell[i]$, as required.

## Conclusion

We have shown that for each $\ell \in \{0, ..., L\}$, constant $c \in \mathrm{con}(D^a_L)$ of level $\ell' \in \{\ell, ..., L\}$, and $i \in \{1, ..., \delta_\ell\}$, we have $\mathbf{v}^c_\ell[i] \leq \bar{\mathbf{v}}^{\pi(c)}_\ell[i]$.

Now let $p$ be the index of $A$ in the canonical encoding. We obtain $\mathbf{v}^a_L[p] \leq \bar{\mathbf{v}}^{\pi(a)}_L[p]$, and thus $\mathbf{v}^a_L[p] \leq \bar{\mathbf{v}}^b_L[p]$, since $b = \pi(a)$ and $a$ is of level $L$. But since $A(a) \in T_\mathcal{M}(D^a_L)$, $\mathbf{v}^a_L[p] \geq t_\mathcal{M}$, thus $\bar{\mathbf{v}}^b_L[p] \geq t_\mathcal{M}$, and finally $A(b) \in T_\mathcal{M}(D_b)$. $\qquad\square$

# B Algorithms

## B.1 Checking Soundness of ELUQ rules

---

**Algorithm 1:** Check the soundness of all rules of the form given in Theorem 3 for MAGNN $\mathcal{M}$

---

**Input** : MAGNN $\mathcal{M}$
**Output:** $S$: set of sound rules for $\mathcal{M}$ of the form given in Theorem 3 that together subsume
      every ELUQ rule that is sound for $\mathcal{M}$

**1** $S \leftarrow \emptyset$                                                             `// result set`
**2 foreach** $n \in \{1, ...\delta + |\mathsf{Col}|\}$ **do**
    `// consider rules whose body contains exactly n concept atoms`
**3**     **foreach** *candidate rule $r$ of form*

$$r : \exists P_1.\top \sqcap \ldots \sqcap \exists P_j.\top \sqcap A_1 \sqcap \ldots \sqcap A_k \sqsubseteq A_{k+1}$$

    *with exactly $n$ body concepts* **do**
        `// check subsumption by a sound (smaller) rule`
**4**         **if** $\exists r' \in S$ *with head $A_{k+1}$ whose body concepts are contained in those of $r$* **then**
**5**             $S \leftarrow S \cup \{r\}$
**6**             **continue**
        `// construct dataset with fresh constants` $a \neq b$
**7**         $D_{\text{base}} \leftarrow \{A_1(a), \ldots, A_k(a), P_1(a,b), \ldots, P_j(a,b)\}$
        `// test the soundness of r, using Proposition 4`
**8**         **if** $A_{k+1}(a) \in T_{\mathcal{M}}(D_{base})$ **then**
**9**             $S \leftarrow S \cup \{r\}$
**10 return** $S$

---

## B.2 Refining Explanations

There are two primary issues with the rules defined in Theorem 6.

First, the size of the rules can be large in practice, making them difficult to read, and also limiting their ability to generalise to other datasets. The size of the rule can be iteratively refined by pruning facts from $D_a$ and then recomputing $C_L^a$.

More precisely, for $D_a' := D_a \setminus \{\alpha\}$ for some unary fact $\alpha \in D_a$ such that $A(a) \in T_{\mathcal{M}}(D_a')$, the rule $r' : C_L^a \sqsubseteq A$ (dependent on $D_a'$) is still sound for $\mathcal{M}$, by the same argument made in Theorem 6. This can be done iteratively, until we find that $A(a) \notin T_{\mathcal{M}}(D_a')$. This works similarly for $D_a' := D_a \setminus D_\alpha$, where $D_\alpha = \{P(c,c_1), ..., P(c,c_n)\}$ for some binary predicate $P$ and constants $c, c_1, ..., c_n$, such that there is no fact $P(c,d) \in D_a$ with $P(c,d) \notin D_\alpha$, and $A(a) \in T_{\mathcal{M}}(D_a')$. Note that there are many different choices for $\alpha$ and $D_\alpha$ in each iteration, so the space should be searched to find explanatory rules with minimal size. Only facts $\alpha$ and sets $D_\alpha$ within the $L$-hop neighbourhood of $a$ should be considered for pruning, as this is what will decrease the number of concepts in the body $C_L^a$.

The second issue is that the inclusion of the $\leq_n P.\top$ term limits their ability to produce facts on other datasets with higher numbers of neighbours. As a possible solution, we can substitute $n$ for some $m > n$. Consider a dataset $D_a$ and fact $A(a) \in T_{\mathcal{M}}(D_a)$. Instead of defining $C_\ell^c := \ldots \sqcap \leq_n P.\top$ in Definition 5, we can define $C_\ell^c := \ldots \sqcap \leq_m P.\top$, for some $m \geq n$. Any $m$ can be chosen such that extending $D_a$ with $D_\ell^c := \{P(c,d_1), ..., P(c,d_{m-n})\}$ for fresh constants $d_1, ..., d_{m-n}$ still leads to $A(a) \in T_{\mathcal{M}}(D_a \cup D_\ell^c)$. If the above is done for multiple constants $c_1, ..., c_k$ and $\ell_1, ..., \ell_k$, it must be ensured that $A(a) \in T_{\mathcal{M}}(D_a \cup D_{\ell_1}^{c_1} \cup ... \cup D_{\ell_k}^{c_k})$.

The simple approach we take in this paper is as follows. Given a predicted fact $A(a)$ on a dataset $D$, we first identify all atoms $A_1, ..., A_n$ such that $A_i(a) \in D$, and check if the rule $A_1 \sqcap ... \sqcap A_n \sqsubseteq A$ is sound using Proposition 4. If it is not, we then identify all binary predicates $P_1, ..., P_m$ such that $P_i(a,b) \in D$ for any constant $b$, and check if the rule $A_1 \sqcap ... \sqcap A_n \sqcap \exists P_1.\top \sqcap ... \sqcap \exists P_m.\top \sqsubseteq A$ is sound using Proposition 4. If either of these rules is sound, it explains the prediction $A(a)$ on $D$.

| Dataset | Agg | Weights | %Acc | %Prec | %Rec | F1 |
|---------|-----|---------|------|-------|------|-----|
| LUBM | Mean | Standard | $97.1 \pm 0.0$ | $96.9 \pm 0.0$ | $97.2 \pm 0.0$ | $97.1 \pm 0.0$ |
|  |  | Non-Neg | $91.5 \pm 0.0$ | $87.8 \pm 0.0$ | $96.4 \pm 0.0$ | $91.9 \pm 0.0$ |
|  | Max | Standard | $95.7 \pm 0.0$ | $95.3 \pm 0.0$ | $96.3 \pm 0.0$ | $95.8 \pm 0.0$ |
|  |  | Non-Neg | $91.6 \pm 0.0$ | $88 \pm 0.0$ | $96.3 \pm 0.0$ | $92 \pm 0.0$ |
|  | Sum | Standard | $96.1 \pm 0.0$ | $95.9 \pm 0.0$ | $96.2 \pm 0.0$ | $96.1 \pm 0.0$ |
|  |  | Non-Neg | $91 \pm 0.0$ | $88.1 \pm 0.0$ | $94.9 \pm 0.0$ | $91.4 \pm 0.0$ |

Table 5: Results for GNNs across various aggregation functions and standard / non-negative weights, for the LUBM dataset. Metrics are computed on the test set and shown with a 95% CI.

| Dataset | Agg | Weights | %Acc | %Prec | %Rec | F1 |
|---------|-----|---------|------|-------|------|-----|
| WN-hier | Mean | Standard | $99.5 \pm 0.0$ | $99.5 \pm 0.0$ | $99.5 \pm 0.0$ | $99.5 \pm 0.0$ |
|  |  | Non-Neg | $99.8 \pm 0.0$ | $99.7 \pm 0.0$ | $99.9 \pm 0.0$ | $99.8 \pm 0.0$ |
|  | Max | Standard | $99.3 \pm 0.0$ | $99.5 \pm 0.0$ | $99.1 \pm 0.0$ | $99.3 \pm 0.0$ |
|  |  | Non-Neg | $99.9 \pm 0.0$ | $99.8 \pm 0.0$ | $100 \pm 0.0$ | $99.9 \pm 0.0$ |
|  | Sum | Standard | $98.9 \pm 0.0$ | $99.5 \pm 0.0$ | $98.3 \pm 0.0$ | $98.9 \pm 0.0$ |
|  |  | Non-Neg | $98.8 \pm 0.0$ | $97.6 \pm 0.0$ | $100 \pm 0.0$ | $98.8 \pm 0.0$ |
| WN-sym | Mean | Standard | $99.4 \pm 0.0$ | $99.5 \pm 0.0$ | $99.3 \pm 0.0$ | $99.4 \pm 0.0$ |
|  |  | Non-Neg | $100 \pm 0.0$ | $100 \pm 0.0$ | $100 \pm 0.0$ | $100 \pm 0.0$ |
|  | Max | Standard | $99.3 \pm 0.0$ | $99.1 \pm 0.0$ | $99.5 \pm 0.0$ | $99.3 \pm 0.0$ |
|  |  | Non-Neg | $100 \pm 0.0$ | $100 \pm 0.0$ | $100 \pm 0.0$ | $100 \pm 0.0$ |
|  | Sum | Standard | $99.1 \pm 0.0$ | $99.3 \pm 0.0$ | $98.8 \pm 0.0$ | $99.1 \pm 0.0$ |
|  |  | Non-Neg | $100 \pm 0.0$ | $100 \pm 0.0$ | $100 \pm 0.0$ | $100 \pm 0.0$ |
| WN-hier_nmhier | Mean | Standard | $86.2 \pm 0.0$ | $84.7 \pm 0.0$ | $88.4 \pm 0.0$ | $86.5 \pm 0.0$ |
|  |  | Non-Neg | $71.6 \pm 0.0$ | $80.1 \pm 0.1$ | $58.8 \pm 0.1$ | $67.2 \pm 0.0$ |
|  | Max | Standard | $86.9 \pm 0.0$ | $85.1 \pm 0.0$ | $89.5 \pm 0.0$ | $87.2 \pm 0.0$ |
|  |  | Non-Neg | $69.5 \pm 0.0$ | $70.1 \pm 0.0$ | $68.4 \pm 0.0$ | $69.2 \pm 0.0$ |
|  | Sum | Standard | $89.5 \pm 0.0$ | $88.1 \pm 0.0$ | $91.4 \pm 0.0$ | $89.7 \pm 0.0$ |
|  |  | Non-Neg | $66.9 \pm 0.0$ | $73.5 \pm 0.1$ | $59.3 \pm 0.2$ | $62.7 \pm 0.1$ |

Table 6: Results for GNNs across various aggregation functions and standard / non-negative weights, for the LogInfer datasets. Metrics are computed on the test set and shown with a 95% CI.

## C    Full Results

Full results for mean, sum, and max aggregation are given in Tables 5 to 7. Randomly sampled sound monotonic rules of the form given in Theorem 3 are shown in Table 8. Randomly sampled explanatory rules for predictions on LUBM of the form given in Theorem 6 are shown in Table 9.

Randomly sampled explanatory rules that have been reduced in size are shown in Table 10. The final entry in the table is an example of a predicted fact where the size reduction process failed and fell back to using Theorem 6.

On average, out of 1990 true positive predictions, the model only failed 22 times to explain the predicted fact using a rule from Theorem 3. For 4 of our 5 trained MAGNNs, it only failed 10 times each. For the other MAGNN, it failed 72 times. One factor that brought up the average number of body concepts is that some of the explanations were very large: for example, the rule that explained the prediction Department(http://www.Department2.University0.edu) had 1914 body concepts.

| Dataset | Agg | Weights | %Acc | %Prec | %Rec | F1 |
|---|---|---|---|---|---|---|
| FB237v1 | Mean | Standard | $68.7 \pm 0.0$ | $95.4 \pm 0.0$ | $39.3 \pm 0.0$ | $55.7 \pm 0.0$ |
| | | Non-Neg | $71.8 \pm 0.0$ | $75.4 \pm 0.0$ | $64.8 \pm 0.0$ | $69.7 \pm 0.0$ |
| | Max | Standard | $67 \pm 0.0$ | $94.9 \pm 0.0$ | $35.9 \pm 0.0$ | $52 \pm 0.0$ |
| | | Non-Neg | $75.8 \pm 0.0$ | $85.9 \pm 0.1$ | $63.4 \pm 0.0$ | $72.6 \pm 0.0$ |
| | Sum | Standard | $68.4 \pm 0.0$ | $99.7 \pm 0.0$ | $36.8 \pm 0.0$ | $53.8 \pm 0.0$ |
| | | Non-Neg | $73.8 \pm 0.0$ | $79.8 \pm 0.1$ | $64.5 \pm 0.0$ | $71.2 \pm 0.0$ |
| WN18RRv1 | Mean | Standard | $93.7 \pm 0.0$ | $98.5 \pm 0.0$ | $88.8 \pm 0.0$ | $93.4 \pm 0.0$ |
| | | Non-Neg | $95.5 \pm 0.0$ | $98.1 \pm 0.0$ | $92.7 \pm 0.0$ | $95.3 \pm 0.0$ |
| | Max | Standard | $93.6 \pm 0.0$ | $97 \pm 0.0$ | $90.1 \pm 0.0$ | $93.4 \pm 0.0$ |
| | | Non-Neg | $95.5 \pm 0.0$ | $98.7 \pm 0.0$ | $92.1 \pm 0.0$ | $95.3 \pm 0.0$ |
| | Sum | Standard | $93.9 \pm 0.0$ | $96.1 \pm 0.0$ | $91.6 \pm 0.0$ | $93.8 \pm 0.0$ |
| | | Non-Neg | $94.9 \pm 0.0$ | $97 \pm 0.0$ | $92.7 \pm 0.0$ | $94.8 \pm 0.0$ |
| NELLv1 | Mean | Standard | $75.2 \pm 0.1$ | $93.8 \pm 0.0$ | $53.4 \pm 0.2$ | $65.7 \pm 0.2$ |
| | | Non-Neg | $93.4 \pm 0.0$ | $88.8 \pm 0.0$ | $99.4 \pm 0.0$ | $93.8 \pm 0.0$ |
| | Max | Standard | $51.2 \pm 0.0$ | $18.8 \pm 0.4$ | $3.5 \pm 0.1$ | $5.9 \pm 0.1$ |
| | | Non-Neg | $93.2 \pm 0.0$ | $92.4 \pm 0.0$ | $94.1 \pm 0.0$ | $93.2 \pm 0.0$ |
| | Sum | Standard | $51.8 \pm 0.0$ | $18.3 \pm 0.4$ | $4.9 \pm 0.1$ | $7.8 \pm 0.2$ |
| | | Non-Neg | $92 \pm 0.0$ | $86.3 \pm 0.0$ | $100 \pm 0.0$ | $92.6 \pm 0.0$ |

Table 7: Results for GNNs across various aggregation functions and standard / non-negative weights, for the standard benchmark datasets. Metrics are computed on the test set and shown with a 95% CI.

| Dataset | Sample Sound Rules |
|---|---|
| LUBM | $\top \sqsubseteq$ Publication
$\top \sqsubseteq$ UndergraduateStudent
$\exists$authorOfPublication.$\top \sqsubseteq$ ResearchAssistant
$\exists$advisorOf.$\top \sqsubseteq$ AssociateProfessor
$\exists$courseTakenBy.$\top \sqcap \exists$hasTeacher.$\top \sqsubseteq$ Course
Department $\sqcap$ UndergraduateStudent $\sqcap \exists$advisorOf.$\top \sqsubseteq$ FullProfessor |
| WN-hier | _member_of_domain_usage$(X,Y) \rightarrow$ _member_meronym$(X,Y)$
_derivationally_related_form$(X,Y) \wedge$ _similar_to$(X,Y) \rightarrow$ _synset_domain_topic_of$(X,Y)$
_also_see$(X,Y) \wedge$ _similar_to$(X,Y) \wedge$ _verb_group$(X,Y) \rightarrow$ _instance_hypernym$(X,Y)$ |
| WN-sym | _has_part$(X,Y) \wedge$ _similar_to$(X,Y) \rightarrow$ _derivationally_related_form$(X,Y)$ |
| WN-hier_nmhier | _has_part$(X,Y) \wedge$ _member_meronym$(X,Y) \rightarrow$ _member_of_domain_usage$(X,Y)$
_has_part$(X,Y) \wedge$ _verb_group$(X,Y) \rightarrow$ _member_of_domain_usage$(X,Y)$
_hypernym$(X,Y) \wedge$ _synset_domain_topic_of$(X,Y) \rightarrow$ _also_see$(X,Y)$ |
| FB237v1 | $\top \rightarrow$ /music/instrument/instrumentalists$(X,Y)$
$\top \rightarrow$ /people/person/sibling_s./people/sibling_relationship/sibling$(X,Y)$
/base/biblioness/bibs_location/state$(X,Y) \rightarrow$ /film/film/genre$(X,Y)$
/olympics/olympic_games/sports$(X,Y) \rightarrow$ /base/popstra/celebrity/dated./base/popstra/dated/participant$(X,Y)$ |
| NELLv1 | organizationterminatedperson$(X,Y) \wedge$topmemberoforganization$(X,Y) \rightarrow$ organizationhiredperson$(X,Y)$
worksfor$(X,Y) \wedge$ organizationterminatedperson$(X,Y) \rightarrow$ organizationhiredperson$(X,Y)$ |

Table 8: Randomly sampled sound monotonic rules of the form given in Theorem 3.

| | |
|---|---|
| Fact | Publication(http://www.Department2.University0.edu/AssociateProfessor7/Publication0) |
| Rule | $\top \sqsubseteq$ Publication |
| Fact | University(http://www.University370.edu) |
| Rule | $\top \sqcap \exists_1$grantedUndergraduateDegreeTo.(GraduateStudent $\sqcap \exists_1$authorOfPublication.( Publication) $\sqcap \leq_1$ authorOfPublication.$\top$) $\sqcap \leq_1$ grantedUndergraduateDegreeTo.$\top \sqsubseteq$ University |
| Fact | GraduateStudent(http://www.Department6.University0.edu/GraduateStudent29) |
| Rule | $\top \sqcap \exists_4$authorOfPublication.($\top, \top$, Publication, Publication) $\sqcap \leq_4$ authorOfPublication.$\top \sqsubseteq$ GraduateStudent |
| Fact | TeachingAssistant(http://www.Department10.University0.edu/GraduateStudent46) |
| Rule | GraduateStudent $\sqcap \exists_3$authorOfPublication.($\top, \top$, Publication) $\sqcap \leq_3$ authorOfPublication.$\top \sqsubseteq$ TeachingAssistant |
| Fact | GraduateStudent(http://www.Department14.University0.edu/GraduateStudent2) |
| Rule | ResearchAssistant $\sqcap \exists_5$authorOfPublication.($\top$, Publication, Publication, Publication, Publication) $\sqcap \leq_5$ authorOfPublication.$\top \sqsubseteq$ GraduateStudent |
| Fact | UndergraduateStudent(http://www.Department5.University0.edu/UndergraduateStudent87) |
| Rule | $\top \sqsubseteq$ UndergraduateStudent |
| Fact | TeachingAssistant(http://www.Department6.University0.edu/GraduateStudent48) |
| Rule | $\top \sqcap \exists_3$authorOfPublication.($\top$, Publication, $\top$) $\sqcap \leq_3$ authorOfPublication.$\top \sqsubseteq$ TeachingAssistant |
| Fact | GraduateStudent(http://www.Department6.University0.edu/GraduateStudent1) |
| Rule | $\top \sqcap \exists_1$authorOfPublication.($\top$) $\sqcap \leq_1$ authorOfPublication.$\top \sqsubseteq$ GraduateStudent |
| Fact | Course(http://www.Department14.University0.edu/Course39) |
| Rule | $\top \sqcap \exists_{14}$courseTakenBy.($\top$, UndergraduateStudent, UndergraduateStudent, UndergraduateStudent, UndergraduateStudent, UndergraduateStudent, UndergraduateStudent, UndergraduateStudent, UndergraduateStudent, $\top, \top$, UndergraduateStudent, UndergraduateStudent, UndergraduateStudent) $\sqcap \leq_{14}$ courseTakenBy.$\top \sqcap \exists_1$hasTeacher.(AssistantProfessor $\sqcap \exists_7$advisorOf.( UndergraduateStudent, UndergraduateStudent, $\top, \top$, TeachingAssistant $\sqcap$ GraduateStudent, GraduateStudent, $\top$) $\sqcap \leq_7$ advisorOf.$\top \sqcap \exists_5$authorOfPublication.(Publication, Publication, Publication, Publication, Publication) $\sqcap \leq_5$ authorOfPublication.$\top$) $\sqcap \leq_1$ hasTeacher.$\top \sqsubseteq$ Course |
| Fact | ResearchAssistant(http://www.Department11.University0.edu/GraduateStudent71) |
| Rule | $\top \sqcap \exists_4$authorOfPublication.(Publication, Publication, Publication, Publication) $\sqcap \leq_4$ authorOfPublication.$\top \sqsubseteq$ ResearchAssistant |
| Fact | Course(http://www.Department14.University0.edu/Course46) |
| Rule | $\top \sqcap \exists_9$courseTakenBy.($\top, \top$, UndergraduateStudent, $\top$, UndergraduateStudent, $\top$, UndergraduateStudent, $\top$, UndergraduateStudent) $\sqcap \leq_9$ courseTakenBy.$\top \sqcap \exists_1$hasTeacher.(Lecturer) $\sqcap \leq_1$ hasTeacher.$\top \sqcap \exists_1$hasTeachingAssistant.(TeachingAssistant $\sqcap$ GraduateStudent $\sqcap \exists_5$authorOfPublication.(Publication, $\top$, Publication, Publication, $\top$) $\sqcap \leq_5$ authorOfPublication.$\top$) $\sqcap \leq_1$ hasTeachingAssistant.$\top \sqsubseteq$ Course |
| Fact | University(http://www.University138.edu) |
| Rule | $\top \sqcap \exists_2$grantedUndergraduateDegreeTo.(GraduateStudent $\sqcap \exists_1$authorOfPublication.( Publication) $\sqcap \leq_1$ authorOfPublication.$\top$, GraduateStudent $\sqcap \exists_5$authorOfPublication.( $\top$, Publication, $\top$, Publication, Publication) $\sqcap \leq_5$ authorOfPublication.$\top$) $\sqcap \leq_2$ grantedUndergraduateDegreeTo.$\top \sqsubseteq$ University |

Table 9: Randomly sampled explanatory rules of the form given in Theorem 6, shown with the corresponding facts produced by the MAGNN.

| | |
|---|---|
| Fact | Publication(http://www.Department10.University0.edu/FullProfessor5/Publication3) |
| Rule | $\top \sqsubseteq$ Publication |
| Fact | UndergraduateStudent(http://www.Department8.University0.edu/UndergraduateStudent170) |
| Rule | $\top \sqsubseteq$ UndergraduateStudent |
| Fact | TeachingAssistant(http://www.Department5.University0.edu/GraduateStudent45) |
| Rule | GraduateStudent $\sqcap \exists$authorOfPublication.$\top \sqsubseteq$ TeachingAssistant |
| Fact | ResearchAssistant(http://www.Department14.University0.edu/GraduateStudent91) |
| Rule | GraduateStudent $\sqcap \exists$authorOfPublication.$\top \sqsubseteq$ ResearchAssistant |
| Fact | ResearchAssistant(http://www.Department1.University0.edu/GraduateStudent14) |
| Rule | GraduateStudent $\sqsubseteq$ ResearchAssistant |
| Fact | ResearchAssistant(http://www.Department4.University0.edu/GraduateStudent26) |
| Rule | $\exists$authorOfPublication.$\top \sqsubseteq$ ResearchAssistant |
| Fact | GraduateStudent(http://www.Department6.University0.edu/GraduateStudent86) |
| Rule | $\exists$authorOfPublication.$\top \sqsubseteq$ GraduateStudent |
| Fact | GraduateStudent(http://www.Department12.University0.edu/GraduateStudent48) |
| Rule | ResearchAssistant $\sqsubseteq$ GraduateStudent |
| Fact | GraduateStudent(http://www.Department3.University0.edu/GraduateStudent44) |
| Rule | ResearchAssistant $\sqcap \exists$authorOfPublication.$\top \sqsubseteq$ GraduateStudent |
| Fact | GraduateStudent(http://www.Department0.University0.edu/GraduateStudent67) |
| Rule | TeachingAssistant $\sqcap \exists$authorOfPublication.$\top \sqsubseteq$ GraduateStudent |
| Fact | GraduateCourse(http://www.Department4.University0.edu/GraduateCourse31) |
| Rule | $\exists$courseTakenBy.$\top \sqcap \exists$hasTeacher.$\top \sqsubseteq$ GraduateCourse |
| Fact | Course(http://www.Department12.University0.edu/Course1) |
| Rule | $\exists$courseTakenBy.$\top \sqcap \exists$hasTeacher.$\top \sqsubseteq$ Course |
| Fact | AssociateProfessor(http://www.Department7.University0.edu/AssociateProfessor4) |
| Rule | $\exists$advisorOf.$\top \sqsubseteq$ AssociateProfessor |
| Fact | AssistantProfessor(http://www.Department13.University0.edu/AssistantProfessor10) |
| Rule | $\exists$advisorOf.$\top \sqsubseteq$ AssistantProfessor |
| Fact | FullProfessor(http://www.Department6.University0.edu/FullProfessor3) |
| Rule | $\exists$advisorOf.$\top \sqsubseteq$ FullProfessor |
| Fact | University(http://www.University772.edu) |
| Rule | $\exists$grantedUndergraduateDegreeTo.$\top \sqsubseteq$ University |
| Fact | University(http://www.University547.edu) |
| Rule | University $\sqcap \exists$grantedUndergraduateDegreeTo.$\top \sqsubseteq$ University |
| Fact | University(http://www.University900.edu) |
| Rule | University $\sqcap \exists_2$grantedDoctoralDegreeTo.($\exists_4$advisorOf.($\top$, GraduateStudent, GraduateStudent $\sqcap$ ResearchAssistant, GraduateStudent $\sqcap$ TeachingAssistant) $\sqcap \leq_4$ advisorOf.$\top \sqcap \exists_{17}$authorOfPublication.($\top$, Publication, Publication, Publication, Publication, Publication, $\top$, $\top$, Publication, Publication, Publication, $\top$, Publication, Publication, Publication, Publication, Publication) $\sqcap \leq_{17}$ authorOfPublication.$\top$, $\exists_{14}$advisorOf.(UndergraduateStudent, UndergraduateStudent, UndergraduateStudent, UndergraduateStudent, $\top$, $\top$, GraduateStudent, GraduateStudent $\sqcap$ ResearchAssistant, GraduateStudent, $\top$, TeachingAssistant $\sqcap$ GraduateStudent, GraduateStudent, GraduateStudent $\sqcap$ TeachingAssistant, GraduateStudent $\sqcap$ TeachingAssistant) $\sqcap \leq_{14}$ advisorOf.$\top \sqcap \exists_9$authorOfPublication.($\top$, Publication, Publication, Publication, Publication, Publication, Publication, $\top$, Publication) $\sqcap \leq_9$ authorOfPublication.$\top$) $\sqcap \leq_2$ grantedDoctoralDegreeTo.$\top \sqcap \exists_1$grantedMastersDegreeTo.(Lecturer) $\sqcap \leq_1$ grantedMastersDegreeTo.$\top \sqsubseteq$ University |

Table 10: Randomly sampled explanatory rules that have been reduced in size, shown with the corresponding facts produced by the MAGNN. The rules are grouped by similar bodies / heads. These rules come from 5 MAGNNs trained with different random seeds.

