# OpenReview forum: "Sound Logical Explanations for Mean Aggregation Graph Neural Networks"
_NeurIPS.cc/2025/Conference — NeurIPS 2025 poster_

### Official Review · Reviewer_1gqb · 2025-06-11

**Clarity:** 1
**Significance:** 2
**Originality:** 2
**Rating:** 4
**Confidence:** 3

**Summary:**

The paper analyzes from a theoretical point of view the problem of providing sound explanations for the restricted class of GNN with non-negative weights and mean aggregation. They prove that it is possible to extract logic formulas to approximate the behavior of the GNN, and carry out experiments to show the interpretability of the resulting formulas.

**Questions:**

Q1: How is your method different from [2,3]?

Q2: How are the predicates reported in Table 3 defined?

**Ethical Concerns:**

["NO or VERY MINOR ethics concerns only"]

**Final Justification:**

During the rebuttal, the authors clarified that some of the points I raised were due to a misunderstanding, which I believe was partially favored by a paper that was difficult to read for a general ML audience (some unintuitive notation, lack of real-world examples to clarify the statements). Presentation issues aside, I believe the results of the paper are interesting and worth publishing, but not enough for a flagship venue like NeurIPS. This is for two main points: the first, the considered GNN family is quite restricted, as also pointed out by other reviewers; The second, the family of formulas that this family can explain soundly is also quite restricted to only monotonic formulas. This restricts quite a bit the potential impact of the work, in my opinion.

**I raised my score from 2 to 3** to reflect the fact that the authors were very proactive and polite in answering all my points.

**EDIT**

Following an additional discussion with authors, I further increased the score from 3 to 4. The authors highlighted some relevant insights in their answers, which I believe can reshape in the good the overall paper. In particular, they motivated well why it can be a sensible design choice to have less expressive models but at the advantage of better interpretability.

**Limitations:**

Yes

**Quality:**

1

**Strengths And Weaknesses:**

**Strengths**

- The problem addressed is highly significant, and providing formal logic-based explanations is a promising research direction

**Weaknesses**

- **W1**: The main weakness is regarding the writing style and clarity of the paper. It is very hard to follow the logical flow, and the fact that the paper introduces an abundant and non-intuitive notation does not make the job easier. Below are some examples that I find particularly representative of this weakeness:

  - Authors refer to GNN with the letter $\mathcal{N}$, which is instead frequently used in previous literature to denote the neighborhood of a node.

  - I could not understand the difference between unary predicates $U$ and node colors $c \in Col$.

  - Line 269 says "we define a dataset $D^a_L$ to be the $L$-hop neighbourhood of the constant a in $D^a$". This is mixing concepts that are apparently distinct, like dataset, constant, and $L$-hop neighborhood.

Overall, I found myself going back several times to the preliminaries to check what symbols mean, and the fact that notation is often introduced as free text in paragraphs makes it very hard to retrieve the desired meaning. A table of symbols might help in this sense, but in general, I recommend a substantial rewriting of the text to favor readability.


- **W2**: The explanations provided in Table 3 are very promising and indicate a very interesting result. However, no details are provided about how those are extracted. Authors refer to Theorem 3, which however does not provide a constructive approach to extract those formulas. Also, it is not clear how predicates are defined, and whether they are user-defined or provided in the dataset. This is a crucial detail, as not every dataset may come with this kind of annotation. The authors should, at least, provide a concise and intuitive algorithm to compute those explanations.


- **W3**: The paper does not compare nor discuss alternative solutions for extracting logic formulas from GNNs already available in the literature, like [1,2,3,4]

[1] Global Explainability of GNNs via Logic Combination of Learned Concepts

[2] Utilizing Description Logics for Global Explanations of Heterogeneous Graph Neural Networks

[3] Logical Distillation of Graph Neural Networks

[4] GraphTrail: Translating GNN Predictions into Human-Interpretable Logical Rules

---

> ### Author Rebuttal · Authors · 2025-07-30
>
> We would like to thank the reviewer for their engagement with our paper and the time they took to review it.
> Please see below for our responses to their comments and questions.
>
> ## Weaknesses
>
> **Query:**
>
> *The main weakness is regarding the writing style and clarity of the paper. It is very hard to follow the logical flow, and the fact that the paper introduces an abundant and non-intuitive notation does not make the job easier.
> Overall, I found myself going back several times to the preliminaries to check what symbols mean, and the fact that notation is often introduced as free text in paragraphs makes it very hard to retrieve the desired meaning. A table of symbols might help in this sense, but in general, I recommend a substantial rewriting of the text to favor readability.*
>
> **Response:**
>
> We appreciate this feedback from the reviewer, and are committed to improving the readability of the paper. We are grateful for the concrete insights that were given on confusing sections of the paper, and would appreciate more of it: this will allow us to adapt the paper to be more readable by the machine learning community at large.
>
> **Query:**
>
> *Authors refer to GNN with the letter $\mathcal{N}$, which is instead frequently used in previous literature to denote the neighborhood of a node.*
>
> **Response:**
>
> This is standard notation in some of the literature, most particularly the works upon which this paper is based [1,2,3]. We typically reserve $G$ for graphs.
>
> **Query:**
>
> *I could not understand the difference between unary predicates $U$ and node colors $c \in \text{Col}$.*
>
> **Response:**
>
> As stated in L68-69, unary predicates are associated with elements of the vector labels of nodes, and edge colours are associated with binary predicates. As shown in L65, the colours correspond to edges, not nodes. We will make this more explicit by editing our paper to state “is a set of directed edges with colour c” (L66).
>
> **Query:**
>
> *Line 269 says ``we define a dataset $D_L^a$ to be the $L$-hop neighbourhood of the constant $a$ in $D^a$''. This is mixing concepts that are apparently distinct, like dataset, constant, and $L$-hop neighborhood.*
>
> **Response:**
>
> As defined in L60-63, a dataset is a set of facts, which contain constants. $L$-hop neighborhood is well-defined in graphs, which makes them well-defined in datasets given the equivalence defined in L112-120. However, we appreciate that the phrase “$L$-hop neighborhood” may be confusing; we will try to find a better term and would welcome any suggestions.
>
> **Query:**
>
> *The explanations provided in Table 3 are very promising and indicate a very interesting result. However, no details are provided about how those are extracted. Authors refer to Theorem 3, which however does not provide a constructive approach to extract those formulas. Also, it is not clear how predicates are defined, and whether they are user-defined or provided in the dataset. This is a crucial detail, as not every dataset may come with this kind of annotation. The authors should, at least, provide a concise and intuitive algorithm to compute those explanations.*
>
> **Response:**
>
> On L310-L313 of our paper, we state how the rules are extracted: specifically, we enumerate all rules of the form given in Theorem 3 up to a number of body concepts, then use Proposition 4 to check for soundness. The latter step is not stated explicitly in this section, it is discussed on L199-L203; to improve readability, we will add a reference to Proposition 4 explicitly on L312. We will also add an algorithm in the appendices to show how the sound rules are computed.
>
> As stated in L58-59, we assume a fixed signature of predicates, thus assuming that they are provided up-front (whether they originate from the dataset or from the user is independent of our claims). In our experiments, we use standard KG benchmarks, all of which come with pre-defined predicates.
>
> **Query:**
>
> *The paper does not compare nor discuss alternative solutions for extracting logic formulas from GNNs already available in the literature.*
>
> **Response:**
>
> We will add comparisons to these papers in our introduction - we thank the reviewer for bringing them to our attention.
>
> ## Questions
>
> **Query:**
>
> *Q1: How is your method different from [4,5]? (please note, we changed the citation numbering)*
>
> **Response:**
>
> With respect to [4], they only consider explanations in EL, which is a fragment of ELUQ. Thus, our work considers more broad sound rules and explanations. Furthermore, the work provides no formal proofs that their rules are sound for the models - they rely only on empirical evidence.
>
> On the other hand, [5] learn decision trees from GNNs, and then extract logical rules from the decision trees. These decision trees are not equivalent to the GNNs they are derived from (as can be seen in the fact that their performance differs from the GNNs they are learned from), so the extracted logical rules are also not strictly sound. Furthermore, it is not clear how understandable / human readable any of these extracted rules are.
>
> In contrast, our work extracts rules directly from the GNN, with the guarantee that they will be sound for the model.
>
> **Query:**
>
> *Q2: How are the predicates reported in Table 3 defined?*
>
> **Response:**
>
> They are defined in the datasets.
>
> ## Citations
>
> [1] David Tena Cucala, Bernardo Cuenca Grau, Boris Motik, and Egor V Kostylev. On the correspondence between monotonic max-sum gnns and datalog. In Proceedings of the International Conference on Principles of Knowledge Representation and Reasoning, volume 19, pages 459 658–667, 2023
>
> [2] David Tena Cucala, Bernardo Cuenca Grau, Egor V Kostylev, and Boris Motik. Explainable gnn based models over knowledge graphs. In International Conference on Learning Representations, 452 2021.
>
> [3] Matthew Morris, David Tena Cucala, Bernardo Cuenca Grau, and Ian Horrocks. Relational graph convolutional networks do not learn sound rules. In Proceedings of the International Conference on Principles of Knowledge Representation and Reasoning, volume 21, pages 897–908, 2024.
>
> [4] Köhler, Dominik, and Stefan Heindorf. "Utilizing Description Logics for Global Explanations of Heterogeneous Graph Neural Networks." CoRR (2024).
>
> [5] Pluska, Alexander, et al. "Logical distillation of graph neural networks." Proceedings of the 21st International Conference on Principles of Knowledge Representation and Reasoning. 2024.

---

> > ### Comment · Reviewer_1gqb · 2025-08-01
> > **Answer to rebuttal**
> >
> > Thank you, authors, for your answer. Below are my follow-ups:
> >
> > ## Notation clarity
> >
> > > This is standard notation in some of the literature
> >
> > I checked two of the cited papers, and I acknowledge the fact that this notation comes from these works. Nonetheless, [1,2,3] all come from the same main authors, and thus, I would not call this subset of literature representative. Of course, I leave to the authors the freedom to choose the notation they think is best for their work. Nonetheless, I just wanted to point out that using unconventional notation makes the paper hard to follow for readers.
> >
> >
> > > colours correspond to edges, not nodes
> >
> > To the best of my knowledge, colors are typically considered to be associated with nodes rather than edges, which are on their side typically referred to as relations. The origin of this comment is rooted in the fact that this may create some confusion for the reader, but I understand that authors are free to choose the nomenclature they find more suitable, and I acknowledge the fact that it is already stated in the paper. Nonetheless, since the amount of notation is relevant, and as already discussed in my review also other notation clashes with the most commonly used in the literature, I would recommend explaining these aspects in greater detail in the final paper.
> >
> >
> > ## Extracting explanations
> >
> >
> > > We enumerate all rules of the form given in Theorem 3, then use Proposition 4 to check for soundness
> >
> > To my understanding, enumerating all rules means to search for any possible subgraph on the input which can be encoded in a manner compatible with rules in Theorem 3, right? Then each candidate rule is tested for soundness, which, to my understanding, means to take this subgraph and feed it back to the GNN that we want to explain.
> >
> > Is this a correct high-level description of the procedure?
> >
> > - If yes, I have two main considerations. First, this procedure is extremely costly. Second, this very much resembles standard post-hoc explainers for GNNs, where they propose candidate subgraphs and score the best subgraph based on how much it emulates the original prediction [GNNExpl, SubgraphX]. Then, I fail to appreciate the novelty of the proposed approach, and I ask the authors to clarify this point.
> >
> > - If no, I apologize for my misunderstanding, and I gently ask the authors to provide a more high-level explanation similar to my attempt.
> >
> >
> >
> > [GNNExpl] GNNExplainer: Generating Explanations for Graph Neural Networks
> >
> > [SubgraphX] On Explainability of Graph Neural Networks via Subgraph Explorations

---

> ### Author Response · Authors · 2025-08-02
> **Response**
>
> Our thanks to the reviewer for their further engagement, particularly with regard to how swift it was. We greatly appreciate the time they are taking and how their feedback is assisting in the improvement of our paper.
> Please see below for our response:
>
> > Nonetheless, [1,2,3] all come from the same main authors, and thus, I would not call this subset of literature representative
>
> Thank you for pointing this out. We concur that our notation does come from a particular subset of the literature, which may not be representative.
> We are committed to making the paper more readable and are not attached to the use of $\mathcal{N}$ for GNNs.
> As such, we are happy to change it to whatever the reviewer would find most intuitive: perhaps simply $f$ or $f_\text{GNN}$?
>
> > To the best of my knowledge, colors are typically considered to be associated with nodes rather than edges, which are on their side typically referred to as relations.
>
> Our thanks also for this. It is true that in the GNN literature, they are typically referred to as "relation types" rather than colours - the language of edge "colouring" comes from graph theory. This is also something we are very happy to change, in the interest of making the paper more readable and accessible to the machine learning community.
>
> > To my understanding, enumerating all rules means to search for any possible subgraph on the input which can be encoded in a manner compatible with rules in Theorem 3, right? Then each candidate rule is tested for soundness, which, to my understanding, means to take this subgraph and feed it back to the GNN that we want to explain.
>
> This is an excellent question and we apologise for the confusion. The reviewer is mistaken about the process: we will try to clarify how it works.
>
> It is true that a candidate rule is checked for soundness by passing a graph through the GNN. However, we do not consider every possible subgraph of the input which can be encoded in a manner compatible with rules in Theorem 3. Instead, we only consider the single dataset (graph) $D_\text{base}$ defined in Proposition 4. $D_\text{base}$ can be thought of as an instantiation of the body of the rule. Thus, for each rule we want to check for soundness, we only require checking the GNN output on a **single graph**.
>
> To clarify further: the number of such rules (and equivalently, the number of such datasets $D_\text{base}$) is on the order of $2^p$, where $p$ is the number of predicates in the signature (i.e. the number of unique unary types and binary relations in the setting). This is independent of the size of the input graph.
>
> The proof sketch given at the bottom of page 5 should hopefully give some ideas as to why this checking procedure is sound.
> Please do let us know if this explanation is still confusing, and we will attempt to clarify further.

---

> > ### Comment · Reviewer_1gqb · 2025-08-04
> > **Response to authors 2**
> >
> > Thank you for your clarifications.
> >
> > > perhaps simply $f$ or $f_\text{GNN}$?
> >
> > I think this is already much more intuitive. Thank you.
> >
> >
> > > edge "colouring" comes from graph theory
> >
> > If you want to keep this nomenclature, mentioning it in the text would already help. Thank you.
> >
> >
> > > Thus, for each rule we want to check for soundness, we only require checking the GNN output on a single graph.
> >
> > Thank you for this clarification. I feel that I have a better understanding now. Although the procedure of checking for soundness is substantially not new, as virtually any explainable technique has to verify that the candidate explanation is a witness of the model's prediction, I think the main novelty comes from defining a family of GNNs where checking for soundness in this way is sound. Also, I think that the approach of extracting explanations is a bit *naive*, in the sense that you are taking any possible candidate formula and checking whether this is entailed by the model. While it is true that this check is sound only for models learning monotonic rules, and this is where the contribution of the paper is about, I believe that, despite being interesting, it is not strong enough to recommend for acceptance at a venue like NeurIPS.
> >
> > I encourage the authors to refactor the paper to make it more understandable to an ML audience and try with another venue, for instance, a more graph-specialized conference like LoG. That said, I'll raise my score from 2 to 3 to reflect the fact that the authors were very proactive and polite in answering all my points. Best of luck!

---

> > > ### Author Response · Authors · 2025-08-05
> > > **Response 2**
> > >
> > > We thank the reviewer for their continued engagement and feedback.
> > >
> > > First, thank you for clarifying your preferences for notation: this is very valuable feedback.
> > >
> > > > I encourage the authors to refactor the paper to make it more understandable to an ML audience and try with another venue, for instance, a more graph-specialized conference like LoG. That said, I'll raise my score from 2 to 3 to reflect the fact that the authors were very proactive and polite in answering all my points. Best of luck!
> > >
> > > We greatly appreciate the reconsideration and the recommendation! Thank you.
> > >
> > > > Although the procedure of checking for soundness is substantially not new, as virtually any explainable technique has to verify that the candidate explanation is a witness of the model's prediction, I think the main novelty comes from defining a family of GNNs where checking for soundness in this way is sound.
> > >
> > > We would like to clarify what we perceive to be some remaining confusion here, which may be impacting the reviewer's perception of the impact / novelty of our paper.
> > >
> > > One of the main thing that sets our work (and other related works, including [1,2,3]) apart from other explainability papers, such as [4,5], is that we go beyond simply guaranteeing that our candidate explanation is a witness of the model's prediction.
> > >
> > > To clarify: in proposition 4, $D_\text{base}$ is more than just a witness of the model's prediction, it is a witness for the *soundness* of the explanatory rule. Soundness is a much stronger notion than witnessing (see the background for the definition). Intuitively, it guarantees that the rule is faithful to the model across *any possible* input. In many other expressivity methods, the witness is only for the particular input that the prediction was generated on, rather than a general rule that holds in every scenario. This is a much stronger and difficult to obtain result.
> > >
> > > Furthermore, we would like to highlight that Theorem 3 is also an MAGNN expressivity result, as we state "this raises concerns for the logical expressivity of MAGNNs, since any sound ELUQ rule is ultimately subsumed by a set of much simpler sound rules."
> > >
> > > [1] David Tena Cucala, Bernardo Cuenca Grau, Boris Motik, and Egor V Kostylev. On the correspondence between monotonic max-sum gnns and datalog. In Proceedings of the International Conference on Principles of Knowledge Representation and Reasoning, volume 19, pages 459 658–667, 2023
> > >
> > > [2] David Tena Cucala, Bernardo Cuenca Grau, Egor V Kostylev, and Boris Motik. Explainable gnn based models over knowledge graphs. In International Conference on Learning Representations, 452 2021.
> > >
> > > [3] Matthew Morris, David Tena Cucala, Bernardo Cuenca Grau, and Ian Horrocks. Relational graph convolutional networks do not learn sound rules. In Proceedings of the International Conference on Principles of Knowledge Representation and Reasoning, volume 21, pages 897–908, 2024.
> > >
> > > [4] Köhler, Dominik, and Stefan Heindorf. "Utilizing Description Logics for Global Explanations of Heterogeneous Graph Neural Networks." CoRR (2024).
> > >
> > > [5] Pluska, Alexander, et al. "Logical distillation of graph neural networks." Proceedings of the 21st International Conference on Principles of Knowledge Representation and Reasoning. 2024.

---

> ### Comment · Reviewer_1gqb · 2025-08-06
> **Answer to Response 2**
>
> Thank you for this additional clarification. I think this discussion is helping me (and hopefully other reviewers) to clarify our minds about this contribution.
>
> I understand that this notion is much stronger than a simple witness of model prediction. But consider now the following statement:
>
> > This raises concerns for the logical expressivity of MAGNNs
>
> This highlights severe limitations of MAGNNs, and I feel this goes against the interests of the authors, as the fact that they can extract sound explanations for GNNs is very much restricted. In a sense, this seems to suggest that the strong notion of explanation soundness they advocate for is actually very much restricted. Then, why would someone be interested in using a MAGNN rather than any other GNN?
>
>
> I have an additional question: Can you please provide me with an example of a sound rule extracted for a MAGNN trained optimally for a task involving counting, like counting whether the input graph has more white than black nodes?

---

> > ### Author Response · Authors · 2025-08-08
> > **Response 3**
> >
> > Our thanks to the reviewer for their continued thorough engagement, and our apologies for the late reply: the main author on the paper fell ill.
> >
> > > This highlights severe limitations of MAGNNs, and I feel this goes against the interests of the authors, as the fact that they can extract sound explanations for GNNs is very much restricted. In a sense, this seems to suggest that the strong notion of explanation soundness they advocate for is actually very much restricted. Then, why would someone be interested in using a MAGNN rather than any other GNN?
> >
> > This is a great and very nuanced question. To clarify: we can check any ELUQ rule for soundness, it's just that any ELUQ rule will be subsumed by sound rules of the form given in Theorem 3.
> > The motivation for using MAGNNs comes from their strong empirical performance, and the fact that mean is a common choice for aggregation function in GNNs.
> >
> > However, it is true that our theoretical results show that MAGNNs have issues from an expressivity perspective: in our opinion, this highlights the importance of theoretical work such as this, as the strong empirical results may obfuscate underlying problems with the model.
> > Furthermore, limits to expressivity can also often be strong inductive biases for a model to have: for example, the fact that GNNs are equivariant / invariant is technically a limit on the types of functions they can express, but it happens to be a very useful inductive bias in many graph problems.
> >
> > > Can you please provide me with an example of a sound rule extracted for a MAGNN trained optimally for a task involving counting, like counting whether the input graph has more white than black nodes?
> >
> > Similarly to what we prove in Proposition 1, an equivalent program for such an MAGNN does not exist, as it goes beyond FO logic. However, some examples for sound rules for an MAGNN that is trained to identify nodes that have more white than black neighbours (note the switch to node classification, as we do not tackle graph classification in our paper) is as follows:
> >
> > $$ \geq_2 \text{Edge} . \texttt{White} ~ \sqcap  \leq_1 \text{Edge} . \texttt{Black} $$
> > $$ \geq_3 \text{Edge} . \texttt{White} ~ \sqcap  \leq_2 \text{Edge} . \texttt{Black} $$
> > $$ \geq_4 \text{Edge} . \texttt{White} ~ \sqcap  \leq_3 \text{Edge} . \texttt{Black} $$
> >
> > ... and so on. The trouble is that to capture the entirety of the MAGNN, you would need infinitely many such rules.

---

> > > ### Comment · Reviewer_1gqb · 2025-08-08
> > > **Response to Authors 3**
> > >
> > > > Furthermore, limits to expressivity can also often be strong inductive biases for a model to have: for example, the fact that GNNs are equivariant / invariant is technically a limit on the types of functions they can express, but it happens to be a very useful inductive bias in many graph problems.
> > >
> > > I agree. In fact, I believe the message of the paper could be strengthened by following this "selling" point. In a sense, you could present MAGNNs as a family of models with limited expressive power -- which is, anyway, shown to have comparable performances in many popular benchmarks -- but with higher interpretability guarantees (I would further discuss the limitation on counting problems for completeness). This would reconcile the two parts of your analysis much better than it is currently done.
> > >
> > > Also, I recently came across the paper below, which, to my understanding, shows that for monotonic formulas, graph explanations are optimally sufficient. I feel this notion is connected with your notion of soundness, and it would be interesting to add a discussion regarding this if you think the connection is there.
> > >
> > > Beyond Topological Self-Explainable GNNs: A Formal Explainability Perspective. ICML 2025
> > >
> > >
> > > Given the additional insights emerged from this additional discussion, I'll further update my score. Good luck!

---

> > > > ### Author Response · Authors · 2025-08-09
> > > > **Response 4**
> > > >
> > > > A final round of thanks from us to the reviewer, most particularly for their good-faith reviewing, the deep initial consideration of our paper, the suggestions for improvement, and their continued efforts to understand our work. Their efforts are greatly appreciated, and we feel that they have gone beyond the call of duty.
> > > >
> > > > > I agree. In fact, I believe the message of the paper could be strengthened by following this "selling" point.
> > > >
> > > > This is an excellent suggestion, thank you. We will adopt it into the central messaging of the paper.
> > > >
> > > > > Also, I recently came across the paper below, which, to my understanding, shows that for monotonic formulas, graph explanations are optimally sufficient. I feel this notion is connected with your notion of soundness, and it would be interesting to add a discussion regarding this if you think the connection is there.
> > > >
> > > > Thank you for sharing this: it looks very promising. We will read it thoroughly and see if it fits with our theories.
> > > >
> > > > > Given the additional insights emerged from this additional discussion, I'll further update my score. Good luck!
> > > >
> > > > Thanks again for your consideration! We cannot see your score on our side anymore (although we can see the other reviewer's scores), but we suppose this to be some kind of error in the system. Perhaps the Area Chair can advise?

---

### Official Review · Reviewer_UPm7 · 2025-06-24

**Clarity:** 3
**Significance:** 2
**Originality:** 2
**Rating:** 3
**Confidence:** 4

**Summary:**

The paper studies the class of logical rules that can be encoded as graph neural networks (GNNs) with mean aggregation and positive weights (MAGNNs). Previous work has focused on sum and max aggregation. The authors propose a class of monotonic rules, expressible in description logic ALCQ, that can be embedded as MAGNNs, and thus can be used as explanations for their behavior. They also show, experimentally, that the restriction to positive weights is not too stringent in practice, and that meaningful explanations can be obtained from practical MAGNNs by using their language.

**Questions:**

- Is there any difference in expressive power between MAPNNs and GNNs with sum aggregation and positive weights?

**Ethical Concerns:**

["NO or VERY MINOR ethics concerns only"]

**Final Justification:**

This is a reasonable paper, but not at the level of NeurIPS. LoG seems like a more suitable venue for these results.

**Limitations:**

Yes, they have

**Paper Formatting Concerns:**

No concerns

**Quality:**

2

**Strengths And Weaknesses:**

Strengths

- The paper is well-written and deals with the interesting and relevant topic of providing explanations to GNNs in the form of logical rules.

Weaknesses:

- The paper feels a bit unambitious, in my view, as theoretical results are a bit shallow and do not establish a clear-cut difference with previous work. For instance, is there any difference between the rules that can be expressed by MAPNNs and GNNs with sum-aggregation and positive weights? The latter can also express properties that lie beyond FO, but I see no reason why the two might not be equivalent in expressive power.

- The fact that ELUQ is the strongest fragment of ALCQ that can be expressed with MAPNNs was a bit disappointing for me. I understand that this class of results is difficult, but certainly not impossible, and I would have appreciated to see at least some effort in this direction.

---

> ### Author Rebuttal · Authors · 2025-07-30
>
> We would like to thank the reviewer for their engagement with our paper and the time they took to review it.
> Please see below for our responses to their comments and questions.
>
> ## Weaknesses
>
> **Query:**
>
> *The paper feels a bit unambitious, in my view, as theoretical results are a bit shallow and do not establish a clear-cut difference with previous work. For instance, is there any difference between the rules that can be expressed by MAPNNs and GNNs with sum-aggregation and positive weights? The latter can also express properties that lie beyond FO, but I see no reason why the two might not be equivalent in expressive power.*
>
> **Response:**
>
> We assume the reviewer to mean “MAGNNs”, instead of “MAPNNs”. GNNs with sum aggregation and positive weights are characterised exactly by sound rules from the language of Datalog with inequalities, as proved by [1]. This means that they cannot express properties that lie beyond FO, since Datalog with inequalities is a fragment of FO (when not computing to a fixpoint). As such, they are strictly not equivalent to MAGNNs when it comes to expressive power, as we show in Proposition 1 (L136) that MAGNNs go beyond FO.
>
> **Query:**
>
> *The fact that ELUQ is the strongest fragment of ALCQ that can be expressed with MAPNNs was a bit disappointing for me. I understand that this class of results is difficult, but certainly not impossible, and I would have appreciated to see at least some effort in this direction.*
>
> **Response:**
>
> We appreciate that it is disappointing that we could not extend our analysis to fragments of ALCQ beyond ELUQ, or ideally to ALCQ itself. The reason for this is that the monotonicity (under injective homomorphisms) of ELUQ is essential for our proofs. Both negation and universal quantification are non-monotonic, so we had to exclude them from our fragment to be able to prove the claims in the paper.
>
> ## Questions
>
> **Query:**
>
> *Is there any difference in expressive power between MAPNNs and GNNs with sum aggregation and positive weights?*
>
> **Response:**
>
> We assume the reviewer to mean “MAGNNs”, instead of “MAPNNs”. GNNs with sum aggregation and positive weights are characterised exactly by sound rules from the language of Datalog with inequalities, as proved by [1]. This means that they cannot express logical functions beyond FO, since Datalog with inequalities is a fragment of FO (when not computing to a fixpoint). As such, they are strictly not equivalent to MAGNNs when it comes to expressive power, as we show in Proposition 1 (L136) that MAGNNs go beyond FO. Furthermore, the class of ELUQ rules that MAGNNs can capture is very limited (given in Theorem 3), whereas sum-GNNs with non-negative weights can capture Datalog rules with inequalities, which is a much richer fragment of FO.
>
> ## Citations
>
> [1] David Tena Cucala, Bernardo Cuenca Grau, Boris Motik, and Egor V Kostylev. On the correspondence between monotonic max-sum gnns and datalog. In Proceedings of the International Conference on Principles of Knowledge Representation and Reasoning, volume 19, pages 459 658–667, 2023

---

### Official Review · Reviewer_Tx4b · 2025-07-02

**Clarity:** 4
**Significance:** 3
**Originality:** 4
**Rating:** 4
**Confidence:** 2

**Summary:**

This paper investigates the expressivity and explainability of Graph Neural Networks (GNNs) using mean aggregation (mean-GNNs), which are popular but lack theoretical analysis. The authors focus on mean-GNNs with non-negative weights (MAGNNs) to isolate non-monotonicity sources.

Theoretical: (a) Characterizes the limited class of monotonic rules (ELUQ) sound for MAGNNs (Theorem 3). (b) Proves MAGNNs can express non-FOL functions (Proposition 1). (c) Defines a restricted FOL fragment to generate sound explanatory rules for any MAGNN prediction (Theorem 6).

Empirical: Shows restricting to non-negative weights preserves performance on benchmarks (Table 1), extracts meaningful/sometimes nonsensical rules (Tables 2–3), and generates instance-specific explanations (Table 8).

**Questions:**

1、	Theorem 3 suggests MAGNNs are limited to trivial monotonic rules. Is this a feature of mean aggregation itself, or an artifact of the non-negative weight constraint?
2、	Why do MAGNNs extract zero sound rules on WN18RRv1 (Table 2) despite decent accuracy (95.5%)? Is this due to theoretical limitations or optimization challenges?
3、	How do you reconcile the 15% F1-score drop for MAGNNs on WN-hier_nmhier (Table 5) with the claim that non-negative weights "do not significantly impact performance"?

**Ethical Concerns:**

["NO or VERY MINOR ethics concerns only"]

**Final Justification:**

The authors have partially addressed my concerns. However, certain statements remain potentially misleading in light of the reported results, and important clarifications are still needed for precision and completeness.

**Limitations:**

Yes

**Quality:**

3

**Strengths And Weaknesses:**

Strengths：
1、	The paper establishes a robust theoretical foundation through meticulous proofs (Theorems 3 & 6, Propositions 1 & 4) while complementing them with well-designed experiments.
2、	This work delivers significant originality as the first formal characterization of the expressivity and explainability of mean-aggregation GNNs.
3、	The paper achieves exemplary clarity through precise definitions and logical theory-experiment separation.

Weaknesses：
1、	Restricting weights to non-negative values simplifies theoretical analysis but severely limits practical applicability. Negative weights are essential for modeling inhibitory relationships.
2、	Rules are trivialized; unclear if results hold for general mean-GNNs. Theorem 3 reduces rules to simplistic forms.

---

> ### Author Rebuttal · Authors · 2025-07-30
>
> We would like to thank the reviewer for their engagement with our paper and the time they took to review it.
> Please see below for our responses to their comments and questions.
>
> ## Weaknesses
>
> **Query:**
>
> *Restricting weights to non-negative values simplifies theoretical analysis but severely limits practical applicability. Negative weights are essential for modeling inhibitory relationships.*
>
> **Response:**
>
> Whilst negative weights are essential for modelling certain types of relationships, past works have found that restricting GNNs to non-negative weights leads to models that are still viable in practice, such as [2] for sum-GNNs and [1] for max GNNs. Our experimental findings also validate that, in practice, mean-GNNs with non-negative weights can still be viable and achieve performance equivalent to (and sometimes even better than) ones with negative weights (see Table 1).
>
> **Query:**
>
> *Rules are trivialized; unclear if results hold for general mean-GNNs. Theorem 3 reduces rules to simplistic forms.*
>
> **Response:**
>
> We consider the fact that Theorem 3 reduces rules to simplistic forms to be a strength of our paper, rather than a weakness. Starting from the general fragment of ELUQ, we can prove that any sound ELUQ rule $r$ is in fact subsumed by much simpler rules. The simpler the rules that subsume $r$, the more powerful the result and the more useful they are for providing explanations.
>
> ## Questions
>
> **Query:**
>
> *Theorem 3 suggests MAGNNs are limited to trivial monotonic rules. Is this a feature of mean aggregation itself, or an artifact of the non-negative weight constraint?*
>
> **Response:**
>
> This is primarily a feature of mean aggregation. The sketch of the proof given in L182 onwards relies on the fact that, as neighbours of the same type are continuously added to a node, the computation at that node will become arbitrarily close to the computation on the graph including only a single neighbour of that type, and no other neighbours. This is similar to the technique used in the proof of Property D.7, of [3], albeit to prove different results, and using weighted mean aggregation.
>
> **Query:**
>
> *Why do MAGNNs extract zero sound rules on WN18RRv1 (Table 2) despite decent accuracy (95.5\%)? Is this due to theoretical limitations or optimization challenges?*
>
> **Response:**
>
> These results show, as stated in the table caption, only the sound rules of the form given in Theorem 3 (which subsume all sound ELUQ rules). We can still use Theorem 6 to derive sound rules that explain predictions, and there may be further sound rules beyond ELUQ, which while sound are difficult (or possibly impossible) to prove to be sound. Ultimately, this is a theoretical limitation of the paper, as we can only check the soundness of certain kinds of rules - the ones given in Proposition 4 and Theorem 6.
>
> **Query:**
>
> *How do you reconcile the 15\% F1-score drop for MAGNNs on WN-hier\_nmhier (Table 5) with the claim that non-negative weights ``do not significantly impact performance''?*
>
> **Response:**
>
> Firstly, this is not a common benchmark, and our claim is that “restricting mean-aggregation GNNs to have non-negative weights does not significantly impact performance on common benchmarks” (L9-10). On the three benchmark datasets (FB237v1, WN18RRv1, NELLv1), performance of non-negative mean-GNNs is comparable or better to those with negative weights. Furthermore, this is one of only a few isolated examples of where performance drops in our experiments, meaning the claim is still broadly true in all our results. Finally, this dataset was proposed by [2] to specifically penalise models that are only able to learn monotonic rules, so it is not surprising that performance decreases when the weights are restricted to only being positive.
>
> ## Citations
>
> [1] David Tena Cucala, Bernardo Cuenca Grau, Egor V Kostylev, and Boris Motik. Explainable gnn based models over knowledge graphs. In International Conference on Learning Representations, 452 2021.
>
> [2] Matthew Morris, David Tena Cucala, Bernardo Cuenca Grau, and Ian Horrocks. Relational graph convolutional networks do not learn sound rules. In Proceedings of the International Conference on Principles of Knowledge Representation and Reasoning, volume 21, pages 897–908, 2024.
>
> [3] Adam-Day, Sam, et al. "Almost surely asymptotically constant graph neural networks." Advances in Neural Information Processing Systems 37 (2024): 124843-124886.

---

> ### Comment · Area_Chair_btZ3 · 2025-08-05
> **Ping**
>
> Dear Reviewer,
>
> The deadline for the author-reviewer discussion is approaching (Aug 8, 11.59pm AoE).
> Please read carefully the authors' rebuttal and engage in meaningful discussion.
>
> Thank you,
> Your AC

---

> > ### Comment · Reviewer_Tx4b · 2025-08-06
> > **Response**
> >
> > Thank you for your feedback, which has partially addressed my concerns.  Nevertheless, the current wording “does not significantly impact performance on common benchmarks” could mislead readers, because Table 5 is part of the main results section and the drop is numerically large. I suggest  rephrasing the claim to “on the three standard inductive benchmarks … performance is comparable or better” . Based on the comprehensive consideration, I keep my rating.

---

> > > ### Author Response · Authors · 2025-08-08
> > > **Response**
> > >
> > > Thanks again to the reviewer.
> > >
> > > We appreciate that the wording “does not significantly impact performance on common benchmarks” could indeed be misleading to readers.
> > > Our thanks to the reviewer for their concrete suggestion for the new wording: we will adopt it into our paper.

---

### Official Review · Reviewer_bVNC · 2025-07-06

**Clarity:** 3
**Significance:** 2
**Originality:** 3
**Rating:** 3
**Confidence:** 3

**Summary:**

The paper presents a theoretical analysis of the expressive capacity and explainability of monotonic mean-aggregating GNNs (MAGNNs). It shows that predictions made by MAGNNs can be explained by combinations of simpler logical rules, and provides a constructive approach to derive such rule sets. In addition, the authors propose a method for mining sound rules from model outputs, enabling post hoc explanation of predictions. These two directions together support the extraction of interpretable rules when applying GNNs to knowledge graphs, contributing to a better understanding of their behavior.

**Questions:**

-	Comparison with rule-mining algorithms: There exist a number of works focusing on rule mining from knowledge graphs, including both deep learning–based and symbolic rule-based approaches (e.g., [1], [2]). To what extent are the rules extracted by these existing methods sound? Do those algorithms—and the proposed method—tend to capture similar types of relational patterns?
-	Non-negative weights: The performance difference between GNNs with standard weights and those constrained to non-negative weights varies across datasets. As I understand it, the non-negative parameter space is a subset of the unconstrained one, yet the model with non-negative weights performs significantly better on the NELL dataset. Could the authors provide an analysis or explanation for this non-trivial behavior?
-	Broader GNN categories: While extending the theoretical framework to other categories of GNN may introduce additional complexity, it would be beneficial to include a discussion or empirical comparison with other GNN variants. Could similar experiments be conducted on more expressive architectures or more complex networks?


Reference:
[1] Xu, Zezhong, et al. "Ruleformer: Context-aware rule mining over knowledge graph." Proceedings of the 29th International Conference on Computational Linguistics. 2022.
[2] Wang, Xiaxia, et al. "Faithful rule extraction for differentiable rule learning models." (2024).

**Ethical Concerns:**

["NO or VERY MINOR ethics concerns only"]

**Limitations:**

Yes

**Quality:**

2

**Strengths And Weaknesses:**

Strengths:
-	The paper provides a formal and clear analysis of the expressivity of MAGNNs, offering theoretical insights into the reasoning strategies employed by GNNs in prediction tasks.
-	Experiments on representative datasets demonstrate the effectiveness of the rule mining algorithm. The extracted rules are interpretable in many cases, which aligns well with the goal of enhancing explainability in knowledge graphs and GNN-based models.
Weaknesses:
-	Both the theoretical analysis and experimental evaluation are limited to mean-GNNs with non-negative weights, which represent a relatively narrow and simplified subset within the broader family of GNN architectures.
-	The experimental section lacks a comparison with existing rule-mining algorithms for knowledge graphs, which makes it difficult to assess the relative effectiveness of the proposed method.
Minor weakness:
-	Some parts of the formalism, particularly those concerning the construction and subsumption of ELUQ rules, are dense and may be difficult to follow without substantial prior knowledge of symbolic reasoning.

---

> ### Author Rebuttal · Authors · 2025-07-30
>
> We would like to thank the reviewer for their engagement with our paper and the time they took to review it.
> Please see below for our responses to their comments and questions.
>
> ## Weaknesses
>
> **Query:**
>
> *Both the theoretical analysis and experimental evaluation are limited to mean-GNNs with non-negative weights, which represent a relatively narrow and simplified subset within the broader family of GNN architectures.*
>
> **Response:**
>
> Whilst the GNN architecture is limited, this was necessary for the analysis and rule extraction we were able to perform, and the results in Table 1 show that this restriction results in models which are still viable in practice. The approach of restriction to non-negative weights has been followed in several past works, for max aggregation [4] and sum aggregation [3]. Our work extends these techniques and analysis to a common aggregation function not yet considered (mean aggregation).
>
> **Query:**
>
> *The experimental section lacks a comparison with existing rule-mining algorithms for knowledge graphs, which makes it difficult to assess the relative effectiveness of the proposed method*
>
> **Response:**
>
> The goal of our paper is not to mine rules from knowledge graphs - this is an added benefit that our results allow for. As stated in our Abstract and Introduction, our motivation is as follows: given that GNNs with mean aggregation are commonly used on knowledge graphs, can we extract sound rules that explain their predictions and determine what kinds of rules can even be sound for such GNNs? With this goal in mind, we believe that comparing against knowledge graph rule-mining approaches would not support the arguments in the paper.
>
> ## Questions
>
> **Query:**
>
> *Comparison with rule-mining algorithms: There exist a number of works focusing on rule mining from knowledge graphs, including both deep learning–based and symbolic rule-based approaches (e.g., [1], [2]). To what extent are the rules extracted by these existing methods sound? Do those algorithms—and the proposed method—tend to capture similar types of relational patterns?*
>
> **Response:**
>
> The rules learned by Ruleformer [1] are given in their section 3, and have the form r (X, Y ) ← r1 (X, Z1) ∧ ... ∧ rT (ZT −1, Y ). This is Datalog (without disjunction, negation, or inequalities), which is equivalent to the rules learned by GNNs with max aggregation and non-negative weights, as proved in [4]. It is unclear whether the extracted Ruleformer rules are sound for the model, as it is not proved in the paper.
>
> By contrast, [2] prove that the rules learned by DRUM (another method) can be unsound or incomplete, and adapt DRUM such that the rules are sound and complete. In the adapted version of DRUM, the sound rules recovered come from Datalog with inequalities. This is equivalent to the rules extracted from GNNs with sum aggregation and non-negative weights, as proved by [3].
>
> The rules we consider are fundamentally different, since mean aggregation introduces non-monotonicity into the model. By contrast, both of the above rule fragments are monotonic under injective homomorphisms.
>
> **Query:**
>
> *Non-negative weights: The performance difference between GNNs with standard weights and those constrained to non-negative weights varies across datasets. As I understand it, the non-negative parameter space is a subset of the unconstrained one, yet the model with non-negative weights performs significantly better on the NELL dataset. Could the authors provide an analysis or explanation for this non-trivial behavior?*
>
> **Response:**
>
> This matches the behaviour seen by [5] for sum-GNNs and [4] for max GNNs (as noted in L321 of our paper). Our hypothesis for this behaviour is that (1) the restriction to non-negative weights acts a regulariser for the model and (2) the patterns in the data do not require negative weights to be captured, so the restriction makes searching the weight space easier for the optimizer. We will add a brief explanation of this to the paper.
>
> **Query:**
>
> *Broader GNN categories: While extending the theoretical framework to other categories of GNN may introduce additional complexity, it would be beneficial to include a discussion or empirical comparison with other GNN variants. Could similar experiments be conducted on more expressive architectures or more complex networks?*
>
> **Response:**
>
> Other works have already performed empirical and theoretical comparisons between GNNs with different aggregation functions, such as [6]. Given that the contribution of our paper is primarily theoretical, we believe that empirical comparisons between the GNNs with mean aggregation used in our paper and other GNN architectures is outside of our scope, and that the wide-spread use of mean-GNNs (see L34 of our paper) is sufficient motivation for the problem.
>
> ## Citations
>
> [1] Xu, Zezhong, et al. "Ruleformer: Context-aware rule mining over knowledge graph." Proceedings of the 29th International Conference on Computational Linguistics. 2022.
>
> [2] Wang, Xiaxia, et al. "Faithful rule extraction for differentiable rule learning models." (2024).
>
> [3] David Tena Cucala, Bernardo Cuenca Grau, Boris Motik, and Egor V Kostylev. On the correspondence between monotonic max-sum gnns and datalog. In Proceedings of the International Conference on Principles of Knowledge Representation and Reasoning, volume 19, pages 459 658–667, 2023
>
> [4] David Tena Cucala, Bernardo Cuenca Grau, Egor V Kostylev, and Boris Motik. Explainable gnn based models over knowledge graphs. In International Conference on Learning Representations, 452 2021.
>
> [5] Matthew Morris, David Tena Cucala, Bernardo Cuenca Grau, and Ian Horrocks. Relational graph convolutional networks do not learn sound rules. In Proceedings of the International Conference on Principles of Knowledge Representation and Reasoning, volume 21, pages 897–908, 2024.
>
> [6] Eran Rosenbluth, Jan Toenshoff, and Martin Grohe. Some might say all you need is sum. In Proceedings of the Thirty-Second International Joint Conference on Artificial Intelligence, pages 4172–4179, 2023.

---

> > ### Comment · Reviewer_bVNC · 2025-08-05
> >
> > Thanks for the authors' response to clarify some important unclear points. While, since no new comparing experiments and results are provided, and i will main my scores.

---

> > > ### Author Response · Authors · 2025-08-05
> > > **Response**
> > >
> > > Our thanks to the reviewer for their response.
> > > However, we hope to have to further engagement with our rebuttal and justification for the following statement by the reviewer:
> > >
> > > > While, since no new comparing experiments and results are provided, and i will main my scores.
> > >
> > > As we stated in response to the reviewer asking about additional experimental results:
> > >
> > > *"The goal of our paper is not to mine rules from knowledge graphs - this is an added benefit that our results allow for. As stated in our Abstract and Introduction, our motivation is as follows: given that GNNs with mean aggregation are commonly used on knowledge graphs, can we extract sound rules that explain their predictions and determine what kinds of rules can even be sound for such GNNs? With this goal in mind, we believe that comparing against knowledge graph rule-mining approaches would not support the arguments in the paper."*
> > >
> > > and also
> > >
> > > *"Other works have already performed empirical and theoretical comparisons between GNNs with different aggregation functions, such as [6]. Given that the contribution of our paper is primarily theoretical, we believe that empirical comparisons between the GNNs with mean aggregation used in our paper and other GNN architectures is outside of our scope, and that the wide-spread use of mean-GNNs (see L34 of our paper) is sufficient motivation for the problem."*
> > >
> > > We would greatly appreciate engagement from the reviewer on these two points, as it seems that there is outstanding disagreement on the necessity of extra experiments for the paper.

---

> ### Comment · Area_Chair_btZ3 · 2025-08-05
> **Ping**
>
> Dear Reviewer,
>
> The deadline for the author-reviewer discussion is approaching (Aug 8, 11.59pm AoE).
> Please read carefully the authors' rebuttal and engage in meaningful discussion.
>
> Thank you,
> Your AC

---

### Decision · Program_Chairs · 2025-09-17

**Decision:**

Accept (poster)

**Comment:**

Reviewers generally appreciated the solid formal analysis but lamented the limited potential impact of the results, which are restricted to a subclass of GNN architectures. The authors reply to reviewer 1gqb does address this specific point, however its effects did not propagate to the other reviewers. More generally, the author-reviewer discussion was useful in that it did have a moderate impact on the reviewer's opinion of the paper.

Keeping all of this into consideration, I am moderately positive about the contribution: notational issues aside, it is clean and also sufficiently interesting/impactful for GNN research.